



# Multi-instrumental analysis of ozone vertical profile and total column in South America: comparison between subtropical and equatorial latitudes

Gabriela Dornelles Bittencourt[1], Damaris Kirsch Pinheiro[2], Hassan Bencherif[3,5], Nelson Begue[3], Lucas Vaz Peres[4], José Valentin Bageston[1], Francisco Raimundo da Silva[6], and Douglas Lima de Bem[2]

[1]National Institute for Space Research, INPE/COESU, Santa Maria, RS, Brazil
[2]Federal University of Santa Maria, Santa Maria, RS, Brazil
[3]Laboratoire de l'Atmosphère et des Cyclones, UMR 8105 CNRS, Université de la Réunion, Reunion Island, France
[4]Federal University of Western Pará, Santarém – PA, Brazil
[5]School of Chemistry and Physics, University of KwaZulu-Natal, Westville, Durban, South Africa
[6]National Institute for Space Research, INPE/COENE, Natal, RN, Brazil

**Correspondence:** Correspondence to: Gabriela Dornelles Bittencourt (gadornellesbittencourt@gmail.com)

**Abstract.** The behavior of ozone gas ($O_3$) in the atmosphere varies according to the region of the globe. Its formation occurs mainly in the equatorial stratospheric layer, through the photodissociation of molecular oxygen with the aid of the incidence of ultraviolet solar radiation, but the highest concentrations of $O_3$ content are found in regions of high latitudes (poles) through large-scale circulation (Brewer-Dobson circulation). This work presents a multi-instrumental analysis in subtropical (in Santa Maria (SM) – 29.4ºS; 53.8ºW) and equatorial (in Natal (NT) - 5.4ºS; 35.4ºW) latitudes of South America, to monitor ozone behavior using $O_3$ vertical profile data (2002 - 2020) and total column ozone data (1979 - 2020). Comparisons between latitudes were also analyzed with data from ozonesondes, which have continuous measurements of the vertical ozone profile through the SHADOZ/NASA network, where there is a reference station in Natal. For this, 19 years of data were analyzed using SABER and SHADOZ data for NT, through the monthly average series of each instrument the monthly climatological behavior for the NT station was identified, analysis of percentage and relative differences that showed a good agreement between both instruments, mainly above 20km altitude. It was possible to identify with the analysis in the lower stratosphere (below 20 km), the ozone content is not correctly represented by the TIMED/SABER satellite. The differences between the latitudes presented interesting analyzes regarding the $O_3$ content in the SM and NT, through the monthly and climatological average of the SABER instrument. Dynamic and photochemical effects can interfere both with $O_3$ formation and its distribution along higher latitudes, through the Brewer Dobson Circulation (BDC). The total column of ozone (TCO) was used, to analyze the main climatic variability that influence the two sites (SM and NT). The data analyzed here to monitor $O_3$ in the atmosphere is available through satellite and ground-based instruments from 1979 to 2020. The instruments showed good agreement between each other (TOMS/OMI x Dobson for Natal, and TOMS/OMI x Brewer for Santa Maria) in the long-term series of $O_3$ content measurements, in line with previous studies for these latitudes in the TCO analysis. For climate variability, wavelet analysis was used over 42 years. The investigation revealed a significant annual cycle in both data series for SM and NT. Other variability pointed out such as the Quasi-Biennial Oscillation (QBO) with significant influences in NT and less significant in SM. In



addition, the solar cycle proved to be important and well established ( 128 months) in both site (subtropical and equatorial). The two locations presented in this work have significant importance in the behavior of ozone due to their latitudinal differences. Furthermore, few studies show this comparison between latitudes in South America using satellite and terrestrial instruments in the analysis of the behavior of $O_3$ gas. The main motivation of this work is to show how this important trace gas behaves in the atmosphere, at different altitudes, latitudes and with different sources of influence, both for $O_3$ vertical profile analysis with satellite data, as well as for TCO.

## 1 Introduction

About 90% of the atmospheric ozone content is found in the stratosphere between 15 and 35 km of altitude (London, 1985; WMO, 1994). It is in this altitude range that the formation of a protective barrier against UV radiation harmful to human health and the biosphere is observed, in the well-known Ozone Layer (Kirchhoff et al., 1996). The abundance of ozone concentration in the stratospheric layer is determined by a combination of dynamic and photochemical processes. Thus, in the high stratosphere (35-50 km) the concentration of $O_3$ is determined by the photochemical production and destruction processes of this gas (Sydney, 1930), while in the low stratosphere (15-30 km) the abundance of $O_3$ is mainly by the dynamic transport processes of this $O_3$ content in the region (Dobson et al., 1930; Brewer, 1949; Bencherif et al., 2007, 2011). The first studies show that the process of ozone formation occurs mainly in the stratosphere at 30 km altitude in a cycle of breaking the molecular oxygen $O_2$ where ultraviolet radiation with wavelengths smaller than 242 nm slowly dissociates the $O_2$, forming ozone gas. This production mechanism first suggested by Sydney (1930), shows the dependence of the latitudinal variation, the solar angle, and the intensity of solar radiation, in the formation of $O_3$. Then, the ozone molecule, formed mainly in the stratospheric region and due to tropical latitudes, is transported to regions of medium and high latitudes through the large-scale Brewer-Dobson circulation (BDC) (Dobson et al., 1930; Brewer, 1949; Butchart, 2014). The variability of this circulation depends on some dynamic parameter, which can significantly influence the behavior of $O_3$ at mid-latitudes. Crutzen (2016) revealed one of the first results to the community regarding the destruction of $O_3$ by the catalytic action of nitrogen oxides, in addition to the catalytic destruction due to man-made chlorofluorocarbons (Molina and Rowland, 1974) introduced in the early 1970s. Several instruments have been used over the years to improve the analysis and understanding of atmospheric ozone. Vertical analysis of ozone content is important because they contribute to understand the behavior of $O_3$ from the troposphere to stratospheric levels. Nowadays, these vertical profiles could be obtained by ozonesondes, balloon borne instruments, and equipment aboard satellites. The Sounding of the Atmosphere using Broadband Emission Radiometry (SABER) instrument, aboard of the TIMED (Thermosphere Ionosphere Mesosphere Energetics Dynamics) satellite (Russell III et al., 1999) provides measurements from 2002 to 2018 and will be used in this study to understand the vertical behavior of $O_3$ in subtropical and tropical latitudes. Many studies present analysis of SABER data to identify the main variability through the vertical behavior of temperature and $O_3$ content (Dutta et al., 2022). Remsberg et al. (2003) showed the behavior of the temperature profiles for February 2002 between 52ºS to 83ºN. They revealed that the dataset is useful for analyzes that relate the dynamics and transport of tracer chemical constituents in the average atmosphere. Nath and Sridharan (2014) identified the main variabilities





and trends in $O_3$ content and temperature behavior using SABER data between 20 - 100 km altitude in the 10 - 15ºN region

between 2002 - 2012. The results showed an intense biannual oscillation during the months of March to May and August to

September (between 25 - 30 km altitude). Joshi et al. (2020) analyzed the seasonal and interannual variability of atmospheric

$O_3$ from 14 years of SABER data (2002-2015) in mid-latitude regions of both hemispheres. The results by Joshi and collab-

orators showed an accumulation of $O_3$ starting in late winter and peaking in early spring in both hemispheres, characterizing

the dynamics of the polar vortex, pre-formation of Antarctic Ozone Hole (AOH). In addition, the annual oscillation stood out

in both hemispheres in the middle atmosphere (between the stratosphere and the lower mesosphere), the half-yearly oscillation

showed peaks between 40-60 km and between 80-100 km of altitude. Ozonesonde measurements proved to be effective in

tropospheric and stratospheric ozone analysis (Thompson et al., 2003; Diab et al., 2004), and the SHADOZ network presents a

wide area of ozonesonde measurements, mainly deployed at tropical latitudes. In this way, the use of satellite instruments helps

to investigate the vertical distribution of $O_3$ in different latitudes of the globe, other than tropical regions, and to characterize

its vertical distribution. Sivakumar et al. (2007) showed measurements of $O_3$ variation and climatology in the stratospheric

layer, on Réunion Island (21.06ºS, 55.31ºE), using in situ measurements of ozonesondes, satellite and the Système D'Analyse

par Observations Zénithales (SAOZ) measurements. Comparative results showed ozone content higher values for ozonesonde

measurements in the lower stratosphere, in addition to a seasonal maximum during spring between 24-28 km height. Due to

the limitation of ozone measurements in SM using ozonesondes, the comparison between ground-based x satellite data was

only for the equatorial latitudes, as the equatorial station has a continuity of sounding measurements since 1998 in Natal, with

the sounding station SHADOZ. Thus, in this work only comparisons between types of measurements of the vertical profile of

the atmosphere for Natal will be presented. Latitudinal differences can modify $O_3$ behavior mainly due to the characteristics

of each region, such as the dynamic effects of each latitude (Reboita et al., 2010). In this way, it is necessary to study different

stations in this work (Santa Maria in the subtropical region of Brazil, and Natal with equatorial latitudes) to compare these

latitudes and understand how ozone behaves with these differences, for this profile data ozone vertical measurement through

satellite measurements and ozonesondes. Analysis of the total column of ozone will also be presented, mainly identifying the

main variability of each latitude in the 42 years. These latitudes have distinct and important characteristics in the dynamics

and long-term maintenance of the $O_3$ content. The first station to be studied is at subtropical latitude in the city of Santa Maria

(29.72ºS; 53.41ºW), where the dynamic influence is important and stands out for the $O_3$ analysis, mainly because it is a region

that has indirect effects of the Antarctic Ozone Hole during its active period in the austral spring. The other station is in the

tropical region of South America, where the city of Natal (5.42ºS; 35.40ºW) was chosen, which is an important region in the

analysis of $O_3$ content due to its location. Natal stands out because it has a station with SHADOZ in situ measurements, where

vertical monitoring of the $O_3$ content is carried out using ozonesondes. These two locations are the most important stations

in Brazil with ground-based ozone measurements which begun on 1992 for SM and 1994 for NT. In this way, it is important

to understand the vertical distribution of ozone in these different regions that stand out for their differences in relation to $O_3$.

To organize this database, the SABER satellite will be validated through measurements of the vertical profile of $O_3$ recorded

at Natal in the framework of the SHADOZ (The Southern Hemisphere ADditional OZonesondes) network. The 19 years of

compared data will be used to show and identify the main characteristics of each region related to the ozone behavior during





this period. TCO measurements from satellite (TOMS/OMI) and ground-based (Brewer/Dobson Spectrophotometer) instruments provide a longer data period, about 42 years at both stations (Santa Maria and Natal), it will be an important database to identify the main climate variability according to the specific characteristics of each latitude. Knowing the importance of monitoring atmospheric $O_3$, this work aims to study the vertical behavior and the total column of $O_3$ in different latitudes of Brazil, one in Santa Maria in the subtropics and in Natal in the equatorial region, using different measurement instruments

such as SABER with data available from 2002 to 2020, and ozonesonde analyzes from the SHADOZ platform for the same data period, while the total column ozone provides about 42 years of data between satellite measurements and ground-based instruments from 1979 to 2020.

## 2 Database and Methodology

### 2.1 Regions of study and data used

This work consists of a multi-instrumental analysis of the vertical profile of stratospheric ozone and TCO to investigate the variability of ozone at different latitudes in Brazil. The Brazilian selected sites were Santa Maria (SM) located in the subtropical region (29.4ºS and 53.7ºW), and Natal (NT) (5.42ºS and 35.40ºW) an equatorial station. SM, is a region which is under influence of strong dynamic leading to the variability of $O_3$ in the region throughout the year. The region has a baroclinic atmosphere and the passage of frontal systems in the region affects the dynamics of winds at high levels of the troposphere,

which are associated with cold fronts and the influence of jet streams (Bukin et al., 2011). These currents are crucial in the vertical distribution of $O_3$, being considered the main tropospheric pattern that is directly linked to the transport of $O_3$ from the stratosphere to the troposphere. The position and speed of these jets (subtropical or polar) as a function of the meridional temperature gradient can determine the variation of ozone in the atmosphere (Reboita et al., 2010; Bukin et al., 2011). These dynamical context in subtropical latitudes have significant relevance in studies with $O_3$, because they have

an indirect relationship with the Antarctic Ozone Hole (AOH) which modifies the $O_3$ content in the region mainly during the austral spring through the ejection of masses from air poor in $O_3$ content from the polar region, where the Antarctic Ozone Hole is located, to regions of subtropical latitudes, as is the case of SM (Kirchhoff et al., 1996; Vaz Peres et al., 2017; Bittencourt et al., 2018, 2019). The equatorial region of Brazil (NT) presents a barotropic atmosphere without a large variation in the temperature gradient. According to Reboita et al. (2010), the Brazilian northeast region located near the equator

experiences different meteorological systems throughout the year influenced by several factors, thus impacting the accumulated precipitation in the place. Such systems have as their main forcing thermodynamic factors diverging from the regions of higher latitudes which observe systems generated from dynamic forcings. During the wet season (from November to March) due to the displacement of the Intertropical Convergence Zone (ITCZ) towards the summer hemisphere, the region observes daily precipitation events. Impact on precipitation such as the circulation of the breeze which helps in the supply of humidity

throughout the region together with the circulation of the South Atlantic High Pressure (ASAS), thus causing radiative heating throughout the day together with the humidity from of these circulations thus favors convection. As it is a region with a strong thermodynamic influence, surface heating can be important in the behavior of ozone, because the region is directly related





to the photochemical production of ozone gas ($O_3$) that forms in the tropical stratosphere around 40 km in altitude. Despite being the gas formation region, equatorial latitudes have the lowest ozone concentration values due to a large-scale meridional

circulation known as the BDC which transports ozone to regions of medium and high latitudes (Brewer, 1949; Dobson, 1968). Some studies show that for the future there is a possible trend of acceleration of the BDC in the lower stratosphere region mainly caused by the increase of greenhouse gases in the atmosphere (Weber et al., 2011; Citaristi, 2022). In addition, NT has one of the oldest SHADOZ stations with data available since 1998 (Thompson et al., 2003; Thompson et al., 2017). This station is used to validate satellite observations from ozonesondes. Figure 1 depicts the map of South America, highlighting

the two stations studied in this work, in SM and NT.

### 2.1.1   Database: SHADOZ Observations

The SHADOZ networks began its activities in 1998 with 14 stations distributed in tropical regions providing measurements of the vertical profile of ozone up to the stratosphere (Thompson et al., 2003a). SHADOZ data is collected by sensors known as electrochemical concentration cells (ECC), where these sensors are attached to a standard radiosonde and released by balloons

to monitor the vertical behavior of ozone between the surface and the stratosphere (Thompson et al., 2007; Witte et al., 2017). In South America, the SHADOZ network has a station in NT, which has been in operation since 1998. Nineteen years of data were analyzed using the new version (V06) available, which offers some variables such as ozone, temperature, pressure, relative humidity, wind direction and speed. Based on this information, in the tropical season of Natal, around 580 daily profiles were identified in the period from 2002 to 2020 with 604 vertical levels (Thompson et al., 2021).

### 140   2.1.2   Database: Vertical Ozone data

The analysis of the vertical profile of $O_3$ in this work is used to investigate the behavior of $O_3$ at different heights and at different latitudes in South America. The analysis here were carried out using measurements from the SABER (Atmosphere Survey using Broadband Emission Radiometry) instrument on board the TIMED (Thermosphere-Ionosphere-Mesosphere Energetics and Dynamics) satellite (Russell III et al., 1999; Dawkins et al., 2018), which provides data from 2002 to 2020. It observes

the Earth in narrow spectral ranges, with high accuracy of carbon dioxide (15 $\mu$m), ozone (9.6 $\mu$m) and nitric oxide (5.3 $\mu$m) emissions (Mlynczak et al., 1993, 2007). The main objective of the SABER experiment is to understand the thermosphere and achieve a major improvement in the understanding of fundamental atmospheric processes (Russell III et al., 1999; Joshi et al., 2020). SABER is one of four experiments on NASA's TIMED mission, launched in May 2000 by a Delta II rocket into a circular orbit of 74.1° ± 0.1° inclined, 625 ± 25 km. In activity from 2002, the SABER satellite provides vertical profile

ozone and temperature data, between 15 and 110 km of altitude. The comparison of these data was necessary to obtain a better accuracy of the instrument in subtropical latitudes, and for this, the SHADOZ station will be used, which also performs measurements of the vertical content of $O_3$, and which has data available since 1998 in the equatorial region of Brazil in Natal. Comparisons were made between vertical profiles (SABER and SHADOZ) for the Natal station, considering the availability of data from the instruments between 2002 and 2020, to carry out a comparison in the equatorial latitudes of Brazil (Witte et al.,

2017, 2018; Thompson et al., 2017; Joshi et al., 2020) with respect to the behavior of $O_3$. The validation of vertical profile





data, such as the SABER data, has already been carried out in other regions by analyzing the behavior of these satellite data
with balloon measurements (ozonesondes) (Thompson et al., 2017; Sivakumar et al., 2007; Toihir et al., 2018), in addition to
analysis with data surface, through TCO (Stauffer et al., 2022). Therefore, one of the objectives of this work is to show the
behavior of ozone vertically in a more detailed comparison in the subtropical latitudes of Brazil, in Santa Maria through data

from the SABER satellite, since this region does not yet have a station with monthly measurements of the ozone content. $O_3$
vertically through the ozonesondes, as is the case in Natal. For this work, the study region was selected using a $\pm 2º$ latitude
and longitude box from the reference points in Santa Maria (SM) and Natal (NT), with a vertical resolution of 0.1 km. During
the 19 years of data, approximately 6.715 daily vertical profiles were identified where the satellite mapped the SM, and 3.681
daily vertical profiles for NT during 2002 – 2020. The profiles were interpolated with a vertical resolution of 0.1 km providing

954 altitude levels in the range of 15 to 110 km in height. The SABER satellite provides data on altitude (in km), latitude
and longitude, temperature (in Kelvin) and ozone mixing ratio - OMR - (in ppmv). Complementary analysis includes monthly
mean and climatology of the ozone vertical profile in the two study regions.

### 2.1.3    Database: Total Column Ozone data

The TCO measurements were also used in this work, as it has 42 years of ozone data available at the two latitudes studied

with data from satellites and ground-based instruments. The period for TCO analysis was between 1979 - 2020 using mainly
ground-based (Brewer Spectrophotometer and Dobson Spectrophotometer) and satellite (TOMS, OMI) instruments. For SM
station, the ground-based instrument used is the Brewer spectrophotometer that performs daily measurements of ultraviolet
radiation, total column ozone (TCO) and sulfur dioxide column ($SO_2$). The beginning of TCO monitoring in SM started with
the Brewer Spectrophotometer model MKIV #081 in the period from 1992 to 1999, while the model MKII #056 operated from

2000 to 2002, and the model MKIII #167 from 2002 to 2017. For NT, the ground-based instrument used was the Dobson Spec-
trophotometer #093. The Dobson instrument is a dual-beam monochromator that measures TCO through absorbed radiation
at two wavelengths (305.5 nm and 325.4 nm) by direct observations of the sun (DS), measured since 1978 at the NT. Satel-
lite instruments for TCO have two main instruments in the period used in this work, the Total Ozone Mapping Spectrometer
(TOMS) and the Ozone Monitoring Instrument (OMI). The TOMS instrument began its activities in 1978, with the launch of

the Nimbus-7 satellite and remained operational until 1994. In 1991, the Meteor-3 satellite was launched and provided TOMS
measurements until December 1994 when it was replaced by the Earth Probe satellite in August 1996 and remained in opera-
tion until December 2006. The TOMS instrument ended its activities and with that the OMI instrument was launched in July
2004 aboard the ERS-2 satellite, in cooperation between the Netherlands Agency for Aerospace Programs (NIVR), the Finnish
Meteorological Institute (FMI) and the National Aeronautics and Space Agency (NASA), providing TCO measurements to

185    this day. The OMI derives from the Total Ozone Mapping Spectrometer (TOMS) instrument of NASA and the European Space
Agency (ESA) Global Ozone Monitoring Experiment (GOME) (onboard the ERS-2 satellite). The new generation of satellite
instruments (OMI) can measure more atmospheric constituents at better terrestrial resolution (13 km x 25 km for OMI vs. 40
km x 320 km for GOME). The average monthly TCO is used for the statistical calculations of the two stations, where the data
series made available by the satellite instruments end up filling gaps in the lack of measurements by ground-based instruments



(Vaz Peres et al., 2017; Sousa et al., 2020). The analysis was separated by times and sets of available instruments, for Santa Maria (TOMS/OMI/BREWER) and Natal (TOMS/OMI/Dobson) both for the period from 1979 to 2020. The series merged ground-based and satellites available is used for TCO statistical calculations and climate variability. In addition, TCO enables a long-term analysis of the $O_3$ content (1979 - 2020) with comparative analysis between the two latitudes of the monthly average, climatological and monthly anomaly, with the objective of identifying the main variability that stand out in the regions, through of wavelet analysis.

## 2.2 Statistics, Comparisons and Variabilities

The investigation of the ozone content at the two selected stations over these 42 years of data, consisting of on-board satellite instruments (TOMS, OMI) and ground-based instruments (Brewer/Dobson Spectrophotometer, Ozonesondes) has distinct particularities that are of paramount importance in monitoring ozone content in tropical and subtropical latitudes. Statistical analysis were performed on the available TCO data to understand this behavior. Comparisons were made between different types of instruments (TOMS/OMI x BREWER; TOMS/OMI x Dobson). Through the monthly average for the 42 years, the following calculations were performed:

*Pearson's correlation coefficient (R):*

$$R = \sum_m \frac{\sum_n (SATELITE - \overline{SATELITE}) - (BREWER - \overline{BREWER})}{\sqrt{\sum_n (SATELITE - \overline{SATELITE}) - (BREWER - \overline{BREWER})}} \tag{1}$$

The $R^2$ that is shown in the results is the value of the squared correlation coefficient. The mean squared error (RMSE), often used to estimate the difference between the values predicted by a model (which in this case is a satellite) and the observed values (Brewer spectrophotometer), also called residuals, aggregates the predictive strength of the variable in a single measurement:

*Mean square error (RMSE):*

$$RMSE = \sqrt{\sum_{i=1}^{n} \frac{(SATELITEi - BREWERi)^2}{n}} \tag{2}$$

The mean bias error (MBE) represents a systematic error in which positive values of MBE represent an overestimation of the data and negative values an underestimation of the observed data in relation to the satellite data:

*Mean bias error (MBE):*

$$MBE = \frac{100}{n} \sum_{i=1}^{n} \frac{SATELITEi - BREWERi}{BREWERi} \tag{3}$$

Statistical calculations provided a more quantitative analysis of each instrument and for each region studied. Regarding the vertical ozone content, the analysis of concurrent profiles was established, that is, profiles identified for the same days of



analysis. From this, the monthly average of the concomitant profiles was used, for each latitude during the period from 2002 to 2020, with data from the SABER satellite for Natal and Santa Maria, and from the SHADOZ data station for NT. The percentage differences were calculated for NT in relation to the SABER monthly average by the SHADOZ monthly average, showing how much in percentage each instrument has in relation to the other, equation 4 shows the calculation that was used:

$$RD(\%) = 100 * (\frac{SABER - SHADOZ}{SHADOZ})$$  (4)

### 2.2.1 Variabilities the Ozone

The identification of the predominant variability of $O_3$ content for the 42 years of available data was studied in this work. The wavelet transforms allow identifying the periodicities that stand out the most in time series analyzed along with their evolution (Torrence and Compo, 1998; Rigozo et al., 2012). Monthly TCO anomalies were used in the wavelet transform method to reveal the main modes of ozone variability (Hadjinicolaou et al., 2005). In this work, the Morlet transformed wavelet consists of a plane wave modulated by a Gaussian function, represented by:

$$\psi_0(\eta) = \pi \frac{1}{4} e^{i\omega_0\eta} e^{\frac{-\eta^2}{2}}$$  (5)

where $\omega_0$ is the non-dimensional frequency; $\eta$ is the non-dimensional time parameter. Considering the discrete time series $(X_n)$, with a fixed time spacing $\Delta t$ and n = 0, ..., N-1, the continuous wavelet transform is in Equation 6:

$$W^2(s) = \sum_{n=0}^{N-1} X_n \psi * [\frac{(n'-n)\Delta t}{s}]$$  (6)

where (*) is the complex conjugate is the period (wavelet scale). The global wavelet spectrum Equation 7 allows to calculate the unbiased estimate of the real power spectrum of the time series, by calculating the average wavelet spectrum over a period.

$$W^2(8) = \sum_{n=0}^{N-1} |W_n(8)|$$  (7)

## 3 Results and Discussion

### 3.1 $O_3$ Vertical Profile

Vertical analysis makes it possible to identify the behavior of the $O_3$ content at different altitudes. For this, data generated by instruments aboard satellites are important to understanding the vertical behavior of some gases in a global capacity. Here we analyzed SABER data for Natal and Santa Maria. First, we compare data from SABER and from balloon soundings for Natal station, a SHADOZ network station. For these analyses we used 19 years (2002-2020) of data for both instruments. Figure 2 presents the data series of the two instruments for Natal (SHADOZ and SABER) focusing on the stratospheric layer. Figure



2a shows the monthly average of SHADOZ vertical profiles between surface up to 30 km altitude, while in 2b the SABER vertical profile ranging from 15 to 50 km altitude, both between the period 2002 to 2020 in $O_3$ mixing ratio (OMR), the blank spaces represent periods without data. Figure 2 shows the maximum mixing ratio of $O_3$ with SHADOZ data are observed mainly between 23 and 30 km of altitude with values varying from 6 to 10 ppmv. Thompson et al. (2017) presented a study comparing available data from the SHADOZ network stations at different latitudes with satellite and surface data at these same stations. The results showed that the city of Natal presented a good agreement between the satellite data and the ozonesondes data, despite the blank data gaps presented during the analysis period. In Figure 2 with SABER data, the highest $O_3$ content is observed in the stratospheric region, between 25-40 km of altitude with values between 6 and 10 ppmv, being more intense in early spring, from late August to late summer, mid-March, and April. Nath and Sridharan (2014) showed that there is a strong biennial trend in relation to the SABER analysis between 10 and 15 ºN, mainly in the stratosphere, in addition to the direct relationship of the quasi-biennial oscillation (QBO) with the $O_3$ variability in these latitudes.

### 3.1.1 SABER x SHADOZ comparisons

Regarding the comparison between the two instruments, the relative differences were calculated, allowing the analysis of how the two databases (satellite and ozone soundings) behave in the tropical region of Brazil. In this way, due to the difference in the measures of the instruments, the analysis of concomitant profiles was carried out, that is, profiles identified for the same days of analysis and the same region, between 15 - 30 km of altitude, through the climatology of each series. The first difference found is in the heights, while SABER provides data between 15 and 110 km in height, data from the SHADOZ network provides measurements ranging from the surface to 30 km in height, approximately where the balloon bursts. Figure 3 shows this comparison in OMR units (ppmv) of the $O_3$ vertical profile from 2002 to 2020 through climatology between 15 and 30 km of altitude. One can observe that the two instruments present similar behavior in almost the entire layer and in the entire period. One point stands out, it is in relation to the comparisons in the initial heights of the analysis, around 15-20 km in height. The most evident differences are in the initial heights in relation to the SABER satellite (black lines). These could be explained mainly by divergences and errors in the boundary region of the satellite measurements (Fig 3). These large differences below 20 km were also identified by Bahramvash Shams et al. (2022) where they identified large percentage differences between reanalysis data (MERRA-2) and satellite profiles (AURA/MLS) and ozonesondes data, which show a good concordance mainly in the stratosphere between 22 - 34 km. The region comprising the upper troposphere and lower stratosphere (UTLS) has a lower sensitivity compared to satellite measurements and this must be considered in comparisons involving low altitudes. However, analyzing figure 3, the most significant differences occur between altitudes between 15 and 20 km in altitude, in the summer months (December, January and February), March and April also present significant differences. In the middle stratosphere, between 20 - 26 km, the two profiles of each instrument (SABER in black and SHADOZ in blue) show a similar behavior and begin to diverge in the upper layer of the stratosphere. Therefore, studies that understand the stratospheric layer using data from the SABER satellites can represent well the behavior of $O_3$, not invalidating the use of satellite data in the analysis of vertical profiles of stratospheric $O_3$. Some studies have already shown the behavior of $O_3$ using SHADOZ and SABER data for different regions of the world. For example, in southern Brazil (SM), Bresciani et al. (2018) performed a



multi-instrument analysis to study an intense ozone depletion event in which the air mass came from the Antarctic Ozone Hole and influenced the region in 2016. The study showed comparisons between satellite data, ground-based instrument, Brewer, and ozonesondes launched to monitor the event. (Toihir et al., 2018) identified long-term $O_3$ variability at eight points across the tropics and subtropics, from January 1998 to December 2012, using TCO data from ground-based instruments, satellites,

and vertical profiles through ozone soundings. The analyzes were carried out at 8 stations in the HS including equatorial, tropical and subtropical latitudes, and the results showed that the main variability that dominates the behavior of $O_3$ in the studied regions is the annual oscillation, with greater influence in the subtropics than in the tropics. Unlike QBO, which is in phase in tropical regions, it has a strong connection with large-scale circulation through the BDC. (Sivakumar et al., 2007) carried out a comparative study at the Irene site (25.9ºS, 28.2ºE), where they analyzed the climatological behavior of $O_3$

between troposphere-stratosphere and its variability through data from ozonesondes SHADOZ and vertical profile of $O_3$ with the AURA/MLS satellite. The results of the comparison of these two instruments showed small differences ranging around $\pm 10\%$ above 20 km where the biggest differences are found with positive differences around 15-20%, which may be related to some error in the satellite measurements MLS compared to SHADOZ measurements where studies show a 5% variation in lower stratosphere averages. From Figure 3, where it was identified that analyzes above 20 km of altitude present many

variations, it was decided to carry out the comparisons by calculating the percentage differences from 20 km of height, and not from 15 km as the data from the SABER satellite offers. Thus, to quantify this comparison in NT, Figure 4 presents the percentage difference in the monthly climatology of the vertical profiles between the two instruments for Natal. This comparison presented in figure 4 showed that, despite some still significant differences around the altitude of 20 km, the rest of the vertical profile proved to be reliable to analyze the vertical atmospheric behavior of the atmosphere, mainly between

22 and 30 km of altitude, in the stratospheric region, showing small differences. As shown in figure 4, the largest percentage differences are still in the months of December and February and being significant on January, March, and April.

### 3.2 $O_3$ vertical profile data in subtropical and tropical latitudes

With the comparison of the TIMED/SABER satellite and SHADOZ data from Natal reveals good agreement between the two data series from an altitude upper to 20 km, the following analysis presents the behavior of ozone in the two latitudes of

this work, Santa Maria representing the subtropics and Natal the equatorial region of Brazil. Figures 5 depicts the monthly average for the 19 years of SABER data available in subtropical latitudes (SM) in blue line and equatorial latitudes (NT) in black line in $O_3$ mixing ratio units. The vertical behavior for SM and NT using the SABER data series is presented in figure 5, where dynamic and photochemical influences modify the behavior of ozone in the different regions analyzed. The analysis of figure 5 shows NT (black) presenting higher ozone values, mainly in the region between 25 and 35 km of altitude

in relation to SM (blue), because there is a higher rate of photochemical ozone production that explains these values higher mixing ratios at equatorial latitudes. The production of $O_3$ is greater in the equatorial region and at approximately 40 km of altitude, and may vary according to latitude, altitude, and season of the year, which shows the dependence of the intensity of solar radiation, zenith angle and variation of latitude in the formation of this gas (WMO, 2007; Seinfeld and Pandis, 2016). However, despite these factors for $O_3$ formation to occur at high altitudes at equatorial latitudes, the highest concentrations




of $O_3$ are found at mid- and high-latitudes, which is explained by the large-scale air movement known as the Brewer-Dobson circulation (Brewer, 1949; Dobson, 1968). The Brewer Dobson circulation transports $O_3$ from low latitudes where it forms to mid and high latitude regions. The balance between the $O_3$ production and destruction processes, in addition to being directly influenced by the amount of sunlight available in each region, also causes variations in the $O_3$ content as the air moves to different locations. Few studies show this difference between subtropical and tropical latitudes in South America in relation

to the vertical variation of $O_3$, as is the case of Sousa et al. (2020) who compared a tropical station (Cachoeira Paulista) and an equatorial station (Natal) using ground-based and satellite data, presenting a trend run analysis for Cachoeira Paulista and NT. The formation of $O_3$ will depend on the latitudinal variation, altitude and season of the year, showing differences in the distribution of $O_3$ from its formation region to regions with the highest concentrations. In spring, total ozone is highest at polar latitudes at around 45°N in the Northern Hemisphere and 45° to 60°S in the Southern Hemisphere, as opposed to winter,

when in equatorial regions the seasonal variation in total ozone is smaller by comparison with polar regions (Holton et al., 1995; Gettelman et al., 2011). These spring $O_3$ maxima are the result of increased ozone transport from the tropics to mid- and high-latitudes during late fall and winter due to the large-scale Brewer-Dobson circulation. This change in large-scale movement can be modified according to different sources of impulse, which cause an acceleration and/or deceleration of the movement of the BDC Neu et al. (2014), modifying the behavior of $O_3$ at different latitudes. The monthly average and the

climatology for the vertical profile data obtained by SABER, figures 6 and 7, show the behavior of $O_3$ at different altitudes: 24 km, 32 km, 40 and 48 km for SM and NT. It is possible to identify particularities that ozone plays at each altitude in the two stations, with NT data in black and SM in blue. At 24 km, the two latitudes are influenced by dynamic processes in the lower stratosphere, although for most of the period the two stations present a quasi-antiphasis behavior, which can be explained by the different dynamic forcings of each region, such as the Quasi-Biennial Oscillation (QBO) showing a direct influence at tropical

latitudes impacting stratospheric ozone chemical and dynamic processes (Anstey and Shepherd, 2014; Naoe et al., 2017). The climatology for the altitude of 24 km (Figure 7) presents a different behavior in the two seasons. In SM, a well-marked annual cycle is observed, with well-defined maximums in winter and minimums in summer months, while in NT the behavior of $O_3$ is more constant, without much variation in its content, which can be explained by the Brewer-Dobson circulation through the impulse atmospheric waves that form in the upper troposphere (Butchart, 2014). At an altitude of 32 km, ozone is influenced by

photochemical processes. Thus, it is possible to observe in figure 6b a greater amount of $O_3$ in the NT, equatorial latitude of the country, different from the SM located in subtropical latitudes. The monthly climatology shows the influence of the seasons, where the maximum values of $O_3$ (9.5 - 12 ppmv for NT, and 8 - 10 ppmv for SM) occur in the hottest seasons (summer) and minimum values of $O_3$ (9 - 11 ppmv for NT and 7 - 9 ppmv for SM) in the coldest seasons (winter). The high altitudes are shown in Figure 7c for 40 km and 7d for 48 km. A biannual cycle for NT and annual cycle for SM could be seen at 40

km, while at 48 km SM also shows a biannual variation of $O_3$, in response to the low $O_3$ concentration at high altitudes for subtropical regions. These variations could be explained mainly by the distribution of $O_3$ from its formation region, in low latitudes, to medium and high latitudes. Sousa et al. (2020) showed that Cachoeira Paulista and NT present maximum values in spring regarding the seasonality of $O_3$ content in both seasons, with a well-established annual cycle, like the results found in this work.





### 3.3 Total Column Ozone in subtropical and equatorial latitudes


Total column ozone analysis was used here for comparison purposes with the vertical profile data, therefore monthly TCO series available for both SM and NT latitudes over the 42-year data period (1979 -2020) are: satellites (TOMS and OMI) and Brewer Spectrophotometer for Santa Maria, and satellites (TOMS and OMI), and Dobson Spectrophotometer for equatorial latitude in Natal. Periods of absence of data in the records of the Brewer/Dobson Spectrophotometer are due to instrument

change and/or some technical problem. In the TOMS satellite records, there is a lack of data between 1994 and 1996, which can be explained by the replacement of satellite instruments from Meteor-3 to Earth Probe. In figures 8a) and 8b) it is possible to observe the daily series of TCO in SM and NT, through the different ground-based and satellite instruments used in the analysis of the behavior of TCO in the subtropical and equatorial region of Brazil. There is a well-defined annual cycle in all data sets presented. In addition, the comparison between the instruments shows a good agreement.Toihir et al. (2018) analyzed satellite

data and ground instrument trends and changes in $O_3$ at tropical and subtropical latitudes in the southern hemisphere. One of their results showed good correlation between the instruments which showed that they were used correctly, mainly satellite data for $O_3$ content analysis. Vaz Peres et al. (2017) for subtropical latitudes and Sousa et al. (2020) for equatorial latitudes, showed a good correlation in TCO data for a shorter period of data, indicating good agreement between satellite and surface area in southern Brazil. Comparison statistical analysis were performed using the difference between surface and satellite

instruments. The SM data series presented (Brewer vs. TOMS) for the period June 1992 to December 2005, containing 2164 pairs of data (daily) and between the period October 2004 to December 2017 (Brewer vs. OMI), with 4621 pairs of data. In SM each dataset represents the monthly series of each instrument, it is observed that the correlation coefficient (R2) with respect to the instruments presented considerably good values, where the values of the correlation coefficient were: 0.88 (BREWER vs TOMS) and 0.92 (BREWER vs OMI). Improvements in satellite equipment over the years may explain this good correlation

between TOMS and OMI. Regarding data from TOMS and OMI satellites, previous studies have shown similar results in relation to comparisons between these TCO measurement instruments over different regions. Antón et al. (2009) compared data from the OMI satellite with different ground-based instruments in the Iberian Peninsula and identified a good correlation between the instruments in the behavior of the TCO. Toihir et al. (2015) analyzed the average monthly behavior of TCO at 13 locations in tropical and subtropical latitudes comparing satellite data from the EUMETSAT program, OMI, and surface

data SAOZ and DOBSON available at these stations, the results found were good correlations between these instruments with values around 0.87. The root means square error (RMSE) showed differences of less than 3% in all data sets, which explains the high correlation values between the instruments, presented above by R2. The mean bias error (MBE) in the analysis of the daily TCO data showed: an overestimation between the TOMS dataset relative to Brewer (0.34), and an underestimation of the OMI satellite data compared to surface instrument data BREWER around -0.07, as shown in Table 1. These results confirm the

effectiveness of TCO measurements at the subtropical station of Santa Maria, during the 42 years of data studied in this work, agreeing with previously presented works that perform similar analysis. Vaz Peres et al. (2017) showed a good correlation in TCO data for a shorter period of data, indicating a good agreement between satellite and surface data in the southern region of Brazil. In this work during the period 1979 - 2020 in NT the coefficient of determination (R2) between Dobson vs TOMS





**Table 1.** Statistical analysis between ground-based instruments (Brewer and Dobson) and satellites (TOMS,OMI) for Santa Maria and Natal between 1979-2020.

| Santa Maria/RS | $R^2$ | RMSE (%) | MBE |
|---|---|---|---|
| Brewer x TOMS | 0.88 | 2.66 | 0.34 |
| Brewer x OMI | 0.92 | 1.96 | -0.07 |
| **Natal/RN** | $R^2$ | RMSE (%) | MBE |
| Dobson x TOMS | 0.87 | 1.49 | -0.41 |
| Dobson x OMI | 0.88 | 1.74 | -0.66 |

was 0.87, and the comparison between Dobson and OMI the R2 found was 0.88. For Natal, the RMSE values were 1.49%
for the Dobson vs TOMS comparison, and 1.74% Dobson vs OMI, which shows a certain agreement with results found by
Sousa et al. (2020) who identified values around 2.81% considering TOMS data and 1.94% with OMI. For the MBE values,
it was observed that both TOMS and OMI underestimate the values of the Dobson Spectrophotometer, presenting negative
values -0.41 (TOMS) and -0.66 (OMI). Table 1 presents a summary of the R2, RMSE and MBE values for comparing the TCO
series in SM and NT. Sousa et al. (2020) presented similar comparison values for the period 1974/1978 – 2013 for equatorial
and tropical latitudes, with R2 values around 0.83 (Dobson vs TOMS) and 0.91 (Dobson vs OMI). These high coefficient of
determination values indicate improvements in the new generation of satellites (the replacement of TOMS by OMI in 2005).

### 3.3.1 TCO Climatology

The monthly climatology and the standard deviation are presented in Figure 9 for the subtropical station in SM (blue) and the
equatorial station in NT (black) for the TCO during the period from 1979 to 2020. Previous works showed, the year variability
stands out in the TCO data in SM, with minimum values in autumn (April and May) between 255 and 260 UD, and maximum
values during the austral spring (September and October) with values between 295 and 300 UD (Vaz Peres et al., 2017;
Bittencourt et al., 2019). For the NT season, it is also possible to identify a well-established annual cycle with minima during
autumn, ranging from 255 DU in May to 280 DU in September/October when it reaches its maximum peak. These results agree
with Sousa et al. (2020) who also identified an annual cycle over the NT equatorial latitude. This variability with minimums
in autumn and maximums during spring is mainly explained by the large-scale movement known as the BDC. This transport
is the dominant process that determines through its meridional circulation that the $O_3$ produced in low latitudes is transported
to medium and high latitudes, causing this maximum to occur during late winter/early spring (London, 1985). Sivakumar et al.
(2007) carried out a study on the climatology and stratospheric ozone variability over the Ile de La Réunion, France (24.06 ºS,
55.3 ºE), for 15 years of data available in satellite instruments (HALOE, SAGE-II, TOMS), ozonesondes, where they were also
identified maximum $O_3$ values in spring and minimum values in autumn. Vaz Peres et al. (2017) identified a similar behavior
in the analysis of $O_3$ content on Santa Maria, at the Southern Space Observatory. (Santos et al., 2016) showed the stratosphere-



troposphere exchanges (STE) for the southern of Brazil, where a greater number of exchanges was identified during the winter and spring months, with a lower frequency during the summer, also explained by the BDC. At the NT equatorial station, comparisons between instruments (Dobson vs TOMS, Dobson vs OMI) were satisfactory for the tropical region, showing that satellite and ground-based measurements agree with previous works. Toihir et al. (2018) analyzed trend and variability of $O_3$ content in 8 locations in equatorial, tropical and subtropical latitudes, comparing data from satellite, ground-based instruments and ozonesondes through the SHADOZ network. The comparison between the instruments for the period from 1998 to 2012 showed good agreement, and in addition SHADOZ vertical profile data were considered of good quality to study and identify the main variables in addition to the trend behavior of the $O_3$ content with this data base. well-established data for the equatorial, tropical and subtropical regions.

### 3.3.2  Variabilities in subtropical and equatorial latitudes

The wavelet transforms allow identifying the behavior of the main periodicities that occur in each region. For SM and NT this analysis was carried out applying the monthly anomaly series of TCO for both stations. Figure 10 presents the set of wavelets for Santa Maria (Figure 10a) and Natal (Figure 10b) for the 42-year period (1979-2020) combining satellite and ground-based data. In all analyses, annual periodicities were removed so as not to mask other important variables in the region. The white contours include regions with a confidence level greater than 95% and the U-shaped curve indicates the cone of influence. In the power spectrum for the two analyzed latitudes, the 11-year solar cycle, in the range of 132 months, stands out as an important variability for subtropical latitudes and in the equatorial region of South America on the $O_3$ content. The Quasi-Biennial Oscillation (QBO) between 16 - 32 months also stood out, mainly for NT that showed a strong influence on the TCO data series in the equatorial latitudes, with greater intensity within the influence cone in the period from 1983 to 1995 and between 2000 and 2015. Naoe et al. (2017) analyzed the future of QBO in ozone in the tropical stratospheric region, where through simulations, related to the increase in the effect gases of the study and the decrease of $O_3$ destroying substances, identified a maximum in the amplitude of QBO between 5-10 hPa suggesting that at this time the photochemistry of the region will depend on temperature to modulate ozone in the tropical stratosphere. Newman (2016) showed an anomalous QBO phase shift over the past 60 years. The anomaly showed a rapid upward shift from the westerly phase of the equatorial winds to an easterly phase between 2015-2016. This sudden QBO phase change resulted in the shortest east phase ever seen in 1953-2016 records. In this way, ozone's response to changes in solar irradiation also plays a potentially important role in climate change, regulating stratospheric temperatures and winds. These changes in the stratosphere can affect tropospheric climate through direct radiative effects and dynamic coupling, which in turn affects patterns of extratropical variability Division (2018). Changes in the Brewer-Dobson Circulation (BDC) also modulate the behavior of ozone in the stratosphere, both by its transport, which is influenced by the strongest and/or weakest impulse in the tropics, and by the chemistry in the formation of $O_3$ at tropical latitudes (WMO, 2023). Despite having a low frequency for subtropical latitudes, this signal was observed at a lower frequency, compared to NT, and in other studies at subtropical latitudes (Rigozo et al., 2012; Toihir et al., 2018). Vaz Peres et al. (2017) presented the TCO monitoring using surface and satellite instruments for Santa Maria from 1992 to 2014, where the main periodicities related to this station were identified Figure 10b shows the behavior of the series of data from the equatorial station of Natal, using TCO





with a combination of satellite data and a Dobson spectrophotometer during the study period (1979 to 2020), it was possible to identify the variability that dominates this region by the wavelet method. Between the period of 16 and 32 months, QBO stands out in the Natal series as one of the main variables for the period. Studies show that QBO is the variability that dominates $O_3$ behavior in tropical latitudes. The equatorial region presents important characteristics in relation to the performance of

the solar cycle and, together with the QBO below 30 hPa, they modulate the wind direction Salby and Callaghan (2000). The QBO, in addition to starting its formation process in the stratospheric equatorial region, dissipates to other latitudes, alternating the wind direction from east to west, thus influencing the behavior of $O_3$ at these latitudes (Baldwin, 2001). The half-yearly and annual variability were removed from the analysis so that other possible variability would not be masked. As previously mentioned, the solar cycle (128 months) proved to be a significant variability in the data series analyzed here, along with the

QBO that had a strong influence on the data series. The variability in $O_3$ content will mainly depend on the time of year and the latitude at which it will be analyzed. The influence of QBO on the behavior of stratospheric ozone is explained by its impact on the chemical and dynamic processes of the gas. Figure 11a for SM and 11b for NT, presents the wavelet analysis for the height of 24 km in the stratospheric layer, through SABER data. The influence of QBO on equatorial latitudes such as NT is evident, different from the subtropical region, which shows a weak influence of QBO on the $O_3$ behavior over the period of

24 km. Vaz Peres et al. (2017) showed that for the SM region an anti-phase is observed between the QBO modulation and the TCO anomaly through surface and satellite instruments in the period 1992-2014. The behavior of $O_3$ in relation to El Niño – Southern Oscillation (ENSO) has little influence on the data analyzed in this work both for TCO and at 24 km of altitude, in relation to subtropical and tropical latitudes. Toihir et al. (2018) analyzed the variability using ozone probe data and identified that the influence of ENSO on the behavior of stratospheric $O_3$ is less than 1%, being more representative in the modulation of

tropospheric $O_3$ (Randel and Thompson, 2011). The variation of the QBO signal mainly in the tropics, where $O_3$ is formed, consists of maximum primary peaks between 7 and 10 hPa in the upper stratosphere and secondary peaks between 20 and 30 hPa Baldwin et al. (2001). In addition, other periodicities can influence the behavior of ozone, even in antiphase, such as ENSO and aerosols, in the release of $SO_2$ to the stratosphere through volcanic activities, among other variables (Division, 2018; Nedoluha et al., 2015) . Seasonal variations can be ignored at these latitudes between the equator and the tropics, as

solar radiation is constant throughout the year (Wakamatsu et al., 1989). Globally, the solar cycle proved to be an important periodicity for the two latitudes presented in this work. Meanwhile, QBO also showed an important significance in the NT data series that stands out in the wavelet analysis in the 24 km stratospheric region, but a less significant intensity in the SM region, where some works show an antiphase of the TCO data in relation to the QBO. A period of data within the 95% reliability axis stands out for SM, ENSO between 64 and 128 months, one between 1985 and 2005. At equatorial latitudes, a significant

influence is observed between ENSO and the Solar Cycle in the period from 1985 to 2015. Studies show that ENSO variability is the dominant mode in the troposphere, where its impact directly affects the tropospheric circulation, in tropical upwelling, causing significant changes in the distribution of $O_3$ content in latitudes such as Natal (Oman et al., 2013), already Bencherif et al. (2020) showed that annual oscillations are the dominant mode both in the TCO and in the tropospheric and stratospheric columns, presenting an important influence on the ozone variability in the city of Irene, South Africa (25.9 ºS, 28.2 ºE) in 20

years of analysis.





## 4 Conclusions

This work describes the behavior of ozone content through a multi-instrumental analysis for two different latitudes in Brazil: Santa Maria (29.4ºS; 53.8ºW) comprising subtropical latitudes, and Natal (5.4ºS; 35.4ºW) representing the equatorial region of Brazil, through the analysis of the vertical profile data in the period of 19 years of data (from 2002 to 2020), and analysis

of the total column ozone (TCO) for the two stations between 1979 to 2020, in order to identify the main climatic variables that influence $O_3$ at both latitudes. Vertical profile data were obtained by the TIMED/SABER satellite and to compare with these data, ozonesondes were also analyzed through the SHADOZ measurement network at the Natal station. For the TCO data series, satellite data (TOMS and OMI) and ground-based data (such as Brewer and Dobson Spectrophotometers) installed and operating in Santa Maria and Natal, respectively, were used. Comparative studies between satellite data (SABER) and

ozonesondes from the SHADOZ network in Natal showed that the greatest relative differences between the two databases are below 20 km of altitude, for the most part. These differences, between 15 and 20 km, can be explained by inconsistencies of the satellites, not representing the UTLS region measurements very well, since the SABER satellite starts it analysis around 15 km of altitude. On the other hand, above 22 km the differences are relatively smaller, varying by less than 20% in most months in the comparison between the two instruments (SABER and SHADOZ), and therefore vertical profiles of the SABER satellite

can represent in a careful way the behavior of $O_3$ for the latitudes that are being studied in this work. These results showed that studies using data from the SABER satellite to analyze the lower stratosphere, below 20 km of altitude, is not a good option for Natal station. Furthermore, there are few studies comparing stations at different latitudes in South America regarding the analysis of vertical $O_3$ content using data provided by satellites. On the other hand, studies that monitor the TCO present good comparative results between different measuring instruments, for different latitudes as well. The latitudinal differences

of the study regions show different behavior of the ozone content. For example, the altitude of 24 km for both Santa Maria and Natal is influenced by dynamic processes in the lower stratosphere region where each region has an important characteristic dynamic forcing in this variation. At altitudes between 32 and 40 km, in the middle and upper stratosphere, the photochemical factor is what controls the $O_3$ content at both latitudes. An annual variation stands out at both latitudes, mainly at 32 km of altitude with maximums during spring/summer and minimums in autumn/winter due to the low incidence of radiation. The

small amount of ozone at altitudes such as 48 km could explained the little variation during the year for subtropical latitudes (Santa Maria). Climate variability for the two latitudes, using TCO data from satellites and ground-based instruments, showed a strong influence of the solar cycle at subtropical and equatorial latitudes. The annual cycle is also a dominant variability in the TCO series in Santa Maria, although it has been removed from the wavelet analyses, the climatology with mixing ratio data from the SABER satellite and from the TCO analysis show this well-defined pattern. At the equatorial latitudes in Natal, in

addition to the solar cycle, the other influence that stands out is the QBO, which modulates the behavior of $O_3$ in the tropical region where the main photochemical processes in the formation of ozone gas occur. Changes in this oscillation (QBO) can interfere with the dynamics of the large-scale movement of the Brewer-Dobson Circulation, modulating the distribution of $O_3$ to other regions.



*Author contributions.* GB, DP, HB, NB and LP designed the methodology and GB, NB, DP, LP performed the analysis. GB, DP, HB, NB,
LP, JB and FS contributed to the discussion of the results. GB, DP, LP and DB prepared the manuscript with contributions from all co-authors.

*Competing interests.* The authors declare that they have no conflict of interest.

*Acknowledgements.* This work is part of the MESO Project "Modelling and forecasting the secondary effects of the Antarctic ozone hole",
registered under no. 8887.130199/2017-00. The authors would like to thank the CAPES (Coordination of Improvement of Higher Education
Personnel) and COFECUB (French Committee for the Evaluation of University Cooperation with Brazil) program responsible for promoting
this research. Thanks also to National Aeronautics and Space Administration for the SHADOZ and SABER data.





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



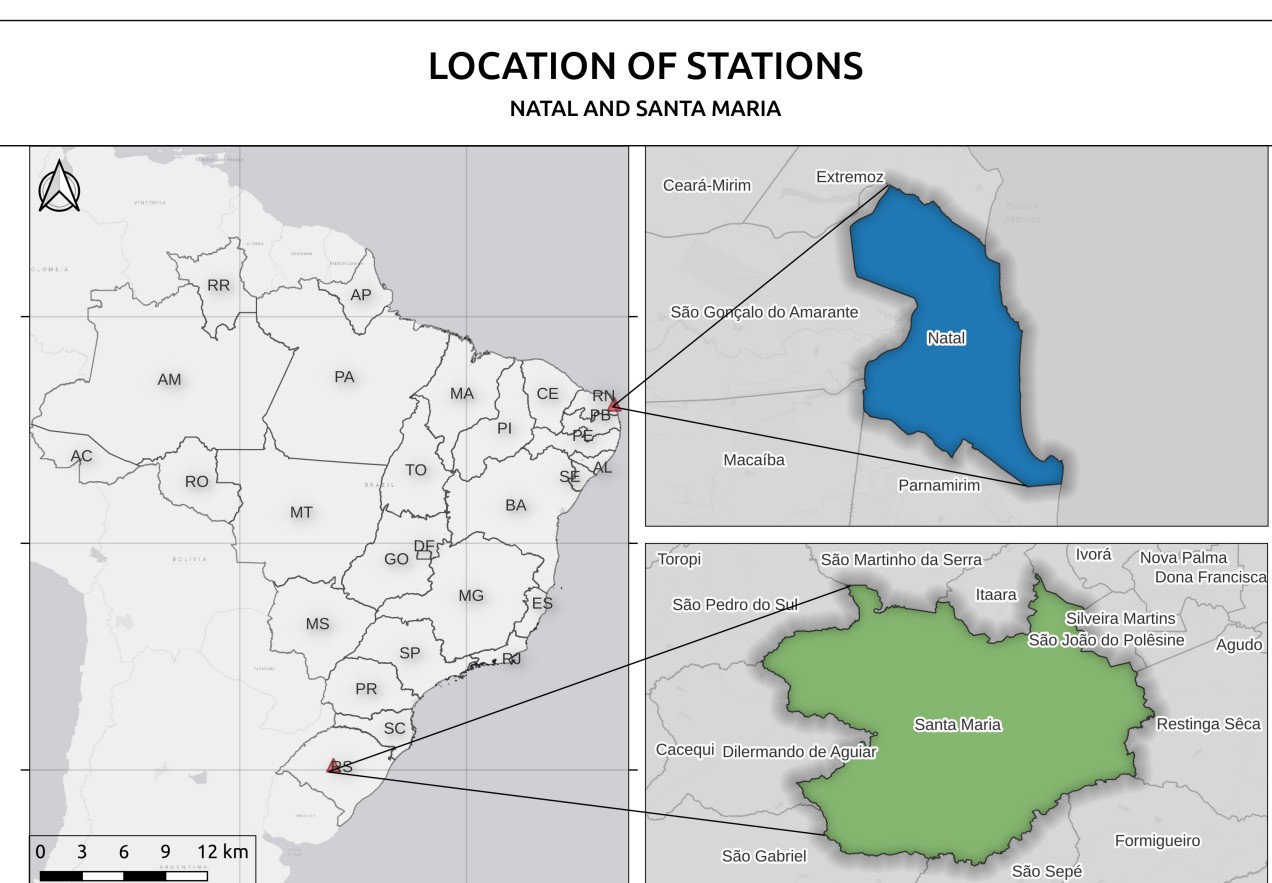

**Figure 1.** Map of South America showing the two stations used in this work, Natal/RN at equatorial latitude and Santa Maria/RS, subtropical region of Brazil.





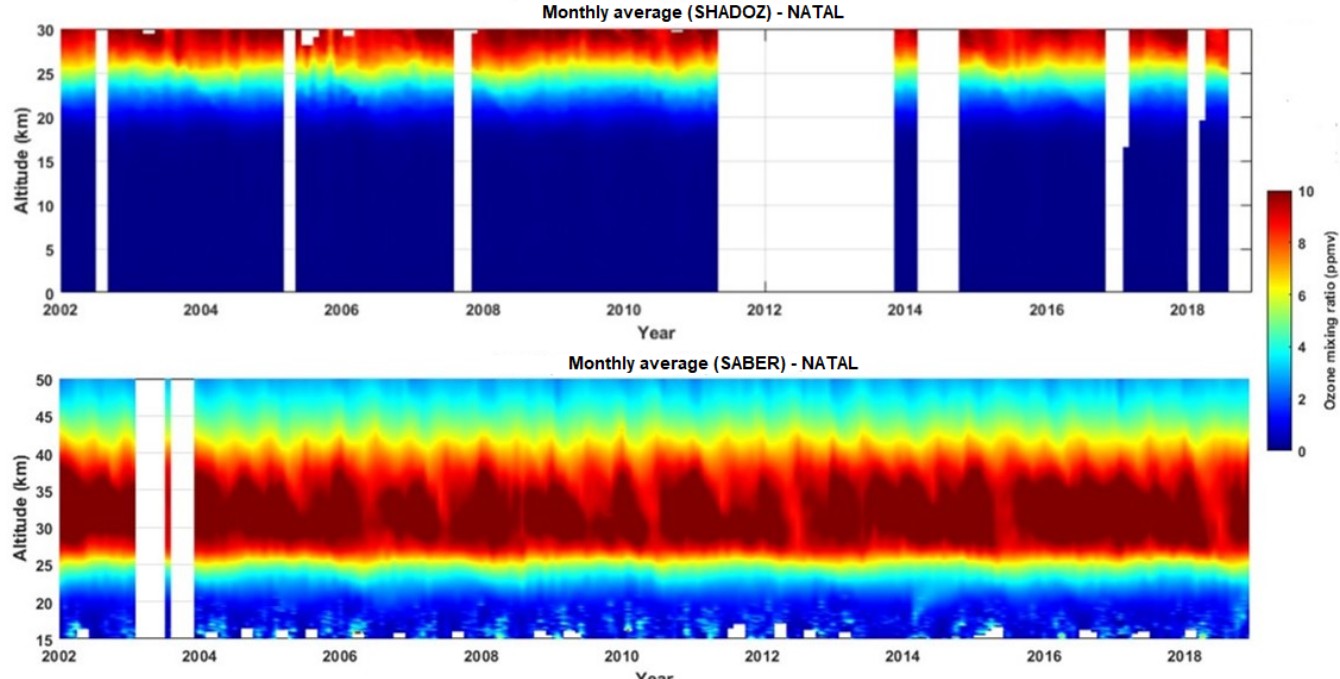

**Figure 2.** Monthly average SHADOZ network, between 0 to 30 km in height, and SABER satellite, between 15-50 km height of the entire data period (2002 - 2020) for Natal.





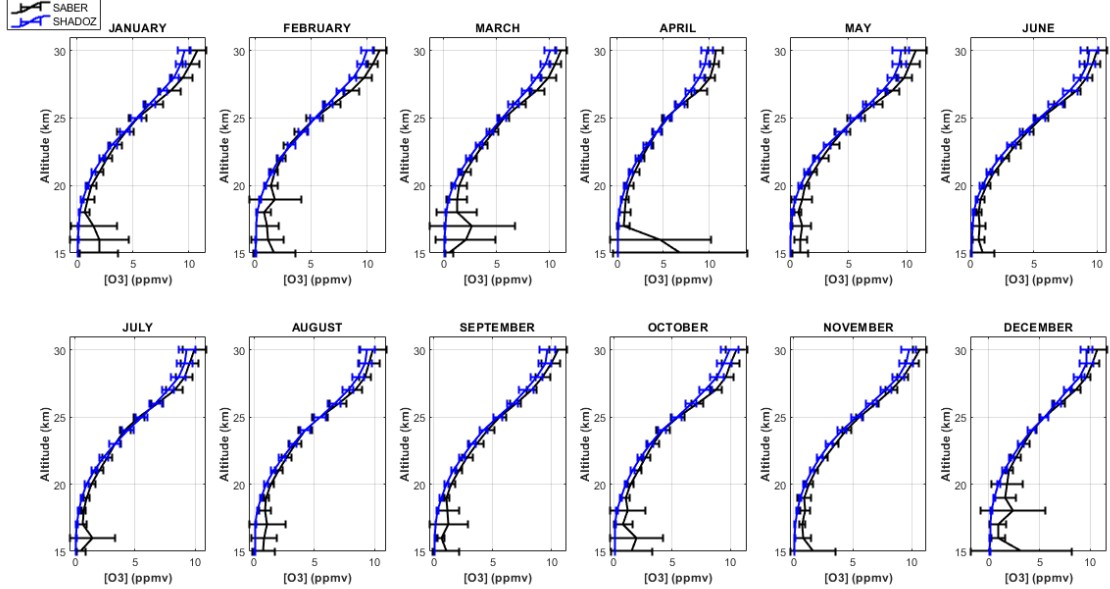

**Figure 3.** Comparison of the vertical profile between SABER (black line) and SHADOZ (blue line), due to $O_3$ mixing ratio (ppmv), of the monthly average of 2002 - 2020 for Natal, between 15 and 30 km of altitude.





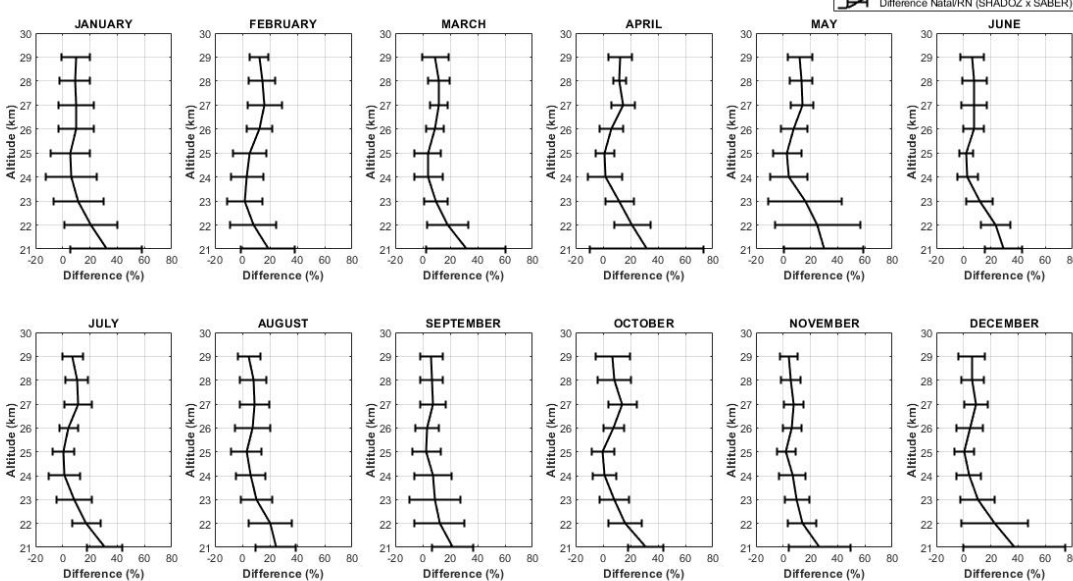

**Figure 4.** Percentage difference in the monthly average between the SHADOZ and SABER instruments for Natal station, from 2002 to 2020.



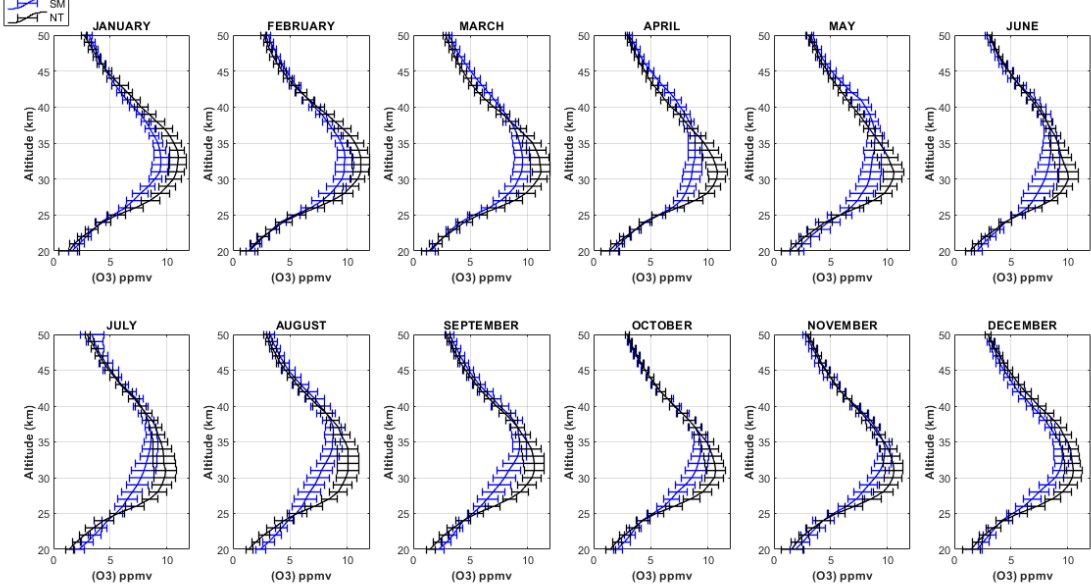

**Figure 5.** Comparison of the vertical profile between Santa Maria (blue) and Natal (black) using SABER data, in $O_3$ mixing ratio units (ppmv) during 2002 – 2020, between 20 and 50 km of altitude.



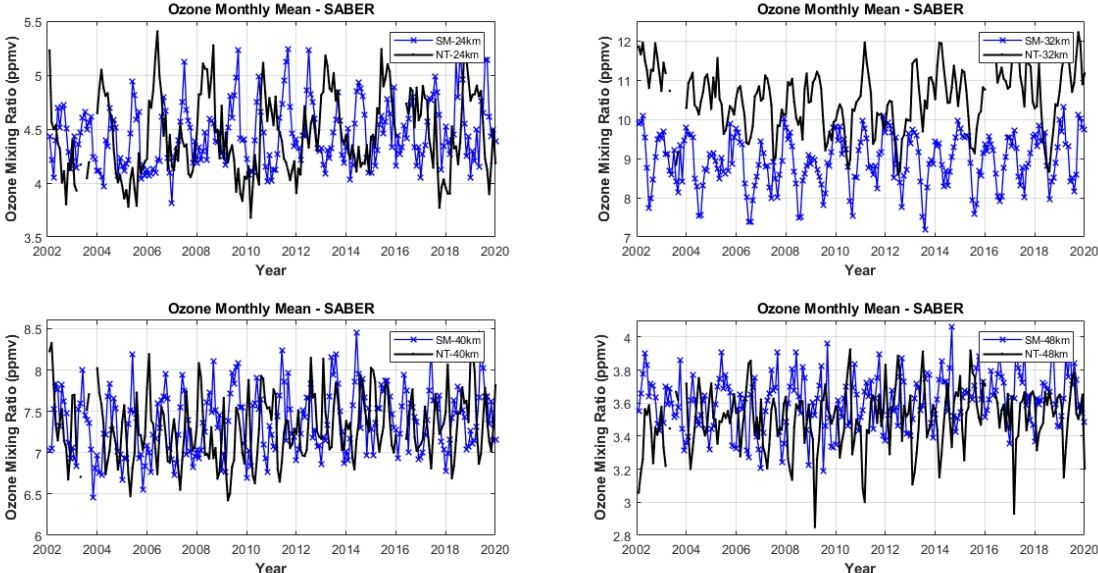

**Figure 6.** Monthly average in 24 km (a), 32 km (b), 40 km (c) and 48 km altitude (d), SABER satellite in Santa Maria (blue) and Natal (black) in ppmv.



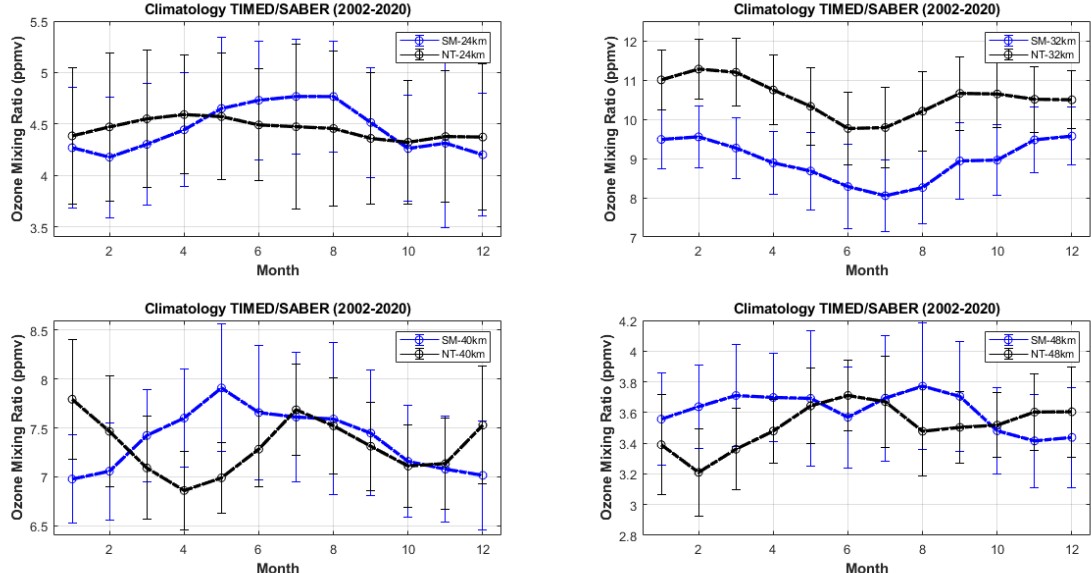

**Figure 7.** Monthly Climatology for SM and NT using SABER data between the period from 2002 to 2020. Differences altitudes, in a) 24 km, b) 32 km, c) 40 km and d) 48 km, blue line is SM and black is NT, in ozone mixing ratio (ppmv).





**Figure 8.** Time series of daily average TCO for each instrument (Brewer, Dobson, and satellite) in Santa Maria (a and b) and Natal (c and d) between 1979 and 2020.



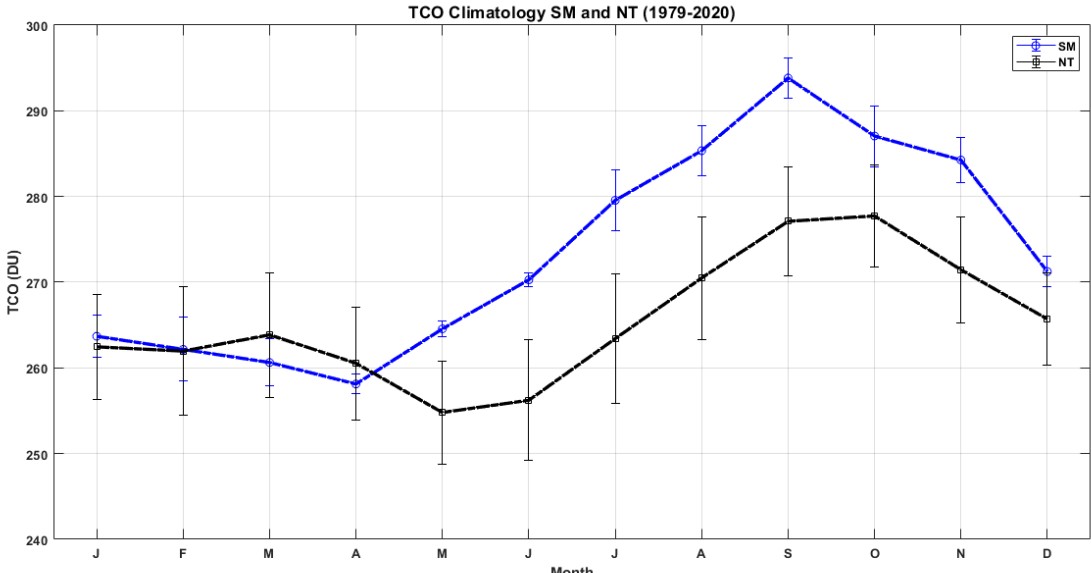

**Figure 9.** Monthly climatology for SM (blue) and NT (black) of TCO in DU between the period 1979 to 2020.



**Figure 10.** Morlet wavelet power spectrum, normalized by $1 / \sigma 2$ with anomaly monthly series of a) Santa Maria and b) Natal using TCO satellite more Brewer/Dobson instrument used for wavelet analysis.





**Figure 11.** Morlet wavelet power spectrum, normalized by 1 / $\sigma2$ with anomaly monthly series in 24 km altitude of a) Santa Maria and b) Natal between 2002 – 2020 in ozone mixing ratio (ppmv) units with SABER satellite instrument used for wavelet analysis.