# Peer review of "Multi-instrumental analysis of ozone vertical profile and total column in South America: comparison between subtropical and equatorial latitudes"

_EGUsphere, 2023_

## Author Comment (AC1)

*We would like to thank the first referee for his time, valuable feedback, and comments. The authors' comments and response (in red), and the changes will be in the new version of the manuscript (in red).*

Review of "Measurement report: Influence of the Antarctic Ozone Hole in Southern Brazil: Conceptual model for 42 years of analysis the atmospheric dynamics on ozone"
Gabriela Dornelles Bittencourt et al.

Unfortunately the resubmitted manuscript did not contain the significant changes I would have expected to see in response to the referees' comments, and therefore very major revisions are still required before this work might be considered acceptabçe for publication.
The authors thank you for your understanding, and a new version of the manuscript will be presented with all the requested explanations and suggestions.

I found it very difficult to follow even the basic steps of the analysis. The writing style is still very unclear. Often a great manu words are used without conveying much information but then the key details are missing. (At least, I couldn't find them).
The text will be remodeled and rewritten for better understanding.

This meant I couldn't really understand what had been done in the different steps in the analysis in sufficient detail and in particular, how the conclusions had been arrived at. This is particularly so for the "conceptual model" which seems to be unconnected to the rest of the manuscript and not supported by the results presented.
More information and details will be included in the new version

I also note that a lot of the analysis conters on the "24.1 – 28 km" height range, even though this is higher up in the stratosphere than the ozone depleted region of the ozone hole (approximately 12-22 km). Perplexingly though, the authors' conceptual model (Figure 14) shows the low-ozone air arriving at the latitude of Santa Maria in the mid-troposphere (700-500 hPa). The SABER data presented in Figures 4, 8 and 9 only goes down to 20 km so it is very hard for me to see how ozone variations in these very different height levels relate to each other (and to variations in total ozone) and how any of the data shown supports the conceptual model.
The initial idea of this manuscript was to present the dynamic behavior of the atmosphere, in mid-latitude regions, with the arrival of events influenced by the Antarctic Ozone Hole, in regions different from its initial position. For this, reanalysis data (ECMWF ERA5) was used with a database of 42 years of data (1979 - 2020), a period that coincides with the total ozone column database for Santa Maria. With this, the objective was to show the stratospheric dynamics, with the potential vorticity fields, and tropospheric dynamics with vertical cutoff fields of the atmosphere in order to identify the behavior of the jet streams, and the possible connection between the entratosphere and the troposphere based on these phenomena of big scale.
Another objective of this work, after identifying the events that influenced mid-latitude regions with the arrival of ozone-poor air masses, was to show the preferred altitude at which these decreases occur for the latitude and longitude of SM. For this objective, data from the SABER satellite was used, which provides 19 years of data (2002 - 2020), where the unit of the vertical ozone profile was in mixing ratio (ppmv). The use of this unit explains the most intense decreases above 30 km, going against what is shown in figure 8, since the mixing ratio depends on the concentration of the $O_3$ content and the concentration of the air.
Although the results differ, the consistency of the satellite vertical profiles shows that the arrival of this poor $O_3$ content in mid-latitudes is around 26 km (according to the figure below in partial pressure of $O_3$). Regarding the model, where only reanalysis data is used, the objective was to show a standard, but not unique, behavior of the atmosphere in the mid-latitude region, specifically Santa Maria, during the active period of the AOH between the months of August to November.

The analysis needs to be made much more coherent and the different sections all clearly linked back to the starting point, which was observed events of low total column ozone.

**Specific comments**

(The text needs a very thorough re-writing to improve the fluency of the English. I have not attempted to list all the places where this is required. Most sentences should be shortened or deleted and the selection of individual words improved).

Lines 13-26 You don't mention SABER in the abstract.
Okay. The abstract was rewritten.

Line 34 Decreasing temperature doesn't "trigger" polar night
Okay. The sentence was rewritten.

Line 44 220 DU is the definition of the ozone hole.
Okay.

Line 91 OMI is not part of the "new generation" in 2023.
Okay. The sentence was rewritten.

Line 127 "Part of the South Pole" – probably you mean Antarctica – the South Pole is a single point.
Okay.

Lines 145-159 This section is very long and repetitive.
Okay. The section has been reorganized.

Line 158-159 You don't say why you've chosen 700 K rather than a lower level in the atmosphere.
Line 158-159 I can't find where you have actually stated the PV criterion for your identification of 'AOH-influenced' events. In line 208 you just say 'Based on the APV examination'

In section 2.1.4 the data used is presented, as well as the reason for analyzing potential vorticity fields, which provide information about the trajectory of trace gases, such as ozone. A decrease (increase) in potential vorticity refers to the polar (equatorial) origin of the air mass, thus, it can be identified whether the event has content from the Antarctic Ozone Hole region for mid-latitude regions.

Line 187 You seem to be saying that, outside the sub-tropics, dynamics only influence ozone at certain times of the year, which is not correct.
The sentence was better rewritten.

Lines 189-191 You seem to be saying that, outside of STE events, meridional temperature gradients do not drive jets.
The sentence was better rewritten.

Lines 190-191 You need to distinguish between ozone variations on the daily timescale, where the position of jets is relevant, and longer-term variations.
The sentence was better rewritten.

Lines 193-195 I don't see the relevance of this work – the seasonal cycle of ozone in the southern hemisphere subtropics has been known since at least the 1960s.
The reference was removed.

Lines 208-209 You don't provide any evidence that your identification method works.
The sentence was better rewritten, as the methodology using PV was described in item 2.1.4.

Line 228 By radiosonde, I assume you mean ozonesonde, but ozonesondes have not been mentioned until this point, and figure 4 shows SABER data.
The sentence was rewritten.

Lines 228-229 An important point that you don't address is that there is a lot of ozone (in Dobson Units) located below 22 km. How do you know the main variation isn't lower down?
Through the analyzes carried out in this work, it was possible to identify that the greatest decreases occur above 22 km, since satellite data below that shows significant differences. The SABER database, during the 19 years analyzed here, showed significant decreases in $O_3$ content between 24.1 and 28 km, considering only visual analyzes of the vertical profiles in the identified events.

Lines 246-255 Is this section needed?
The paragraph was rewritten and reorganized.

Lines 256-270 Is this section needed? Perhaps all this background is intended to support your use of PV to identify Antarctic influence? If so, you should be more explicit.
The paragraph was rewritten and reorganized.

Lines 271-284 In my opinion, this whole section is hampered by the discussion being in terms of features which exist in the climatological mean but not necessarily on a single daily slice of the real atmosphere, which is always more complicated and doesn't necessarily fit into the idealized concepts. In figure 6A, very strong westerlies are obviously present in the stratosphere at about 40 degrees south. I don't think your statements can be justified.
The authors thank you for the suggestion. The paragraph was remodeled for better clarity of the facts. The objective is to show that these AOH-influenced events reach mid-latitude regions, going against what is expected in climatology. In figure 6, which represents the behavior of the atmosphere in a vertical section at the longitude of SM, the connection between the stratospheric and tropospheric jet streams is observed, helping to transport this content from below $O_3$, from the AOH region in Antarctica to mid latitudes, such as SM.

Line 313 I thought you had identified your events by the combination of low total ozone and PV (from reanalysis), not by "ground-based and satellite-based instruments"?
This preliminary analysis of the events of decrease in the total $O_3$ content is carried out by first analyzing the TCO database (ground-based – Brewer Spectrophotometer, and satellite – TOMS and OMI), according to figure 2 which presents the monthly climatology from TCO to SM with the different databases, and according to equation 1. With the "possible days of events" identified, the PV analysis is carried out (reanalysis data – ECMWF ERA5) to identify the origin of the mass ozone-poor air. In short, the satellite data, only from TCO and not the vertical profile data, serves to complement the Brewer database, to obtain the most continuous basis possible.

Lines 320-323 You need to check whether the reductions in ozone mixing ratio between 24 and 28 km add up to decrease seen in total ozone. 28 km is quite high.
As mentioned above, vertical profile analyzes by SABER were presented in $O_3$ mixing ratio (which considers air concentrations and $O_3$ content). The figure below shows the monthly vertical profile between 2002-2020 in SM between 24.1-28 km altitude, with the average number of events identified per month, in relation to the monthly climatology for the 19 years in black, in partial pressure units of $O_3$ (μhPa).

[Figure]

Lines 323-329 This discussion is much too vague. You need to be more specific about the variability at different height levels at the latitude of Santa Maria. Depleted ozone in the Antarctic ozone hole is below 20 km.

The paragraph has been rewritten.

Lines 330 You need to explain your method more clearly.

According to the histogram shown in figure 7, the altitude groups were separated to better visualize these events. During the 19 years analyzed, and according to the events identified and presented in table 2, 43 event profiles were analyzed. Visually, these profiles showed a different behavior in each event, and thus groups were established to identify the greatest decreases, above 30%, in relation to the climatology that was established with the data used. Thus, the "group" with the greatest decreases was between 24.1 – 28 km, with a peak preferably at 26 km.

Line 352 The implication is Bittencourt 2022 first documented the fact the polar vortex "reaches its maximum intensity in later winter and early spring".

The reference was misplaced in this sentence and will be removed in the new version of the manuscript.

Line 370-373 Now you are talking about stratosphere-troposphere exchange, but in the previous section it was all about ozone above 20 km. This section seems completely irrelevant to everything up to this point. The paper seems to change direction very abruptly.

The sentence was removed.

Lines 380-445 The first two paragraphs consist of background and then some very general statements. The "conceptual model" is introduced in line 396 but without any justification, I can't see that you have really provided any support for it at all. At the start of the fourth paragraph, you talk about filaments of low ozone coming away from the vortex – this is what is suggested by the PV maps you have shown such as Figures 5, 10 and 11 – however I can't see that you have made any connection between the PV at 700 K and your conceptual model.

The section has been rewritten, with a more concise and clear description.

Lines 415-452 The conclusion is mostly a repetition of older results followed by a leap to the "conceptual model", again without any real support as far as I can see.

The conclusion was rewritten in the new version of the manuscript.

Figures 1-5 are clear but appear to be based on previous work that has already been published.
They present the initial representation of the work with a new reanalysis database (ECMWF ERA5), for example, in addition to a longer period of data than other works published to date.

Figure 5 shows PV on the 700 K isentropic surface but confusingly, Figure 6 doesn't even reach 700 K and the focus of the discussion is on tropospheric jets. The figures don't seem to be connected to each other.
Figure 5 shows the stratospheric dynamic behavior through the potential vorticity fields, at 700 K potential temperature. While figure 6 shows the dynamic behavior of the atmosphere, through the vertical cut at the longitude of SM (53.7ºW), where stratospheric jets are observed - polar vortex (between 100 to 5 hPa) and tropospheric jets - polar and subtropical jet (around 200 – 250 hPa) in the purple shaded regions, decreasing the wind speed in the color scale. They complement each other as they show the behavior of the atmosphere on the day of the recorded event, according to the methodology applied.

I can't understand what has been plotted in Figure 7 – how have these 'events' been determined? Is there a separate criterion for each height range?
The objective of figure 7 is to show a simple histogram, separated by groups of vertical altitudes analyzed through SABER data in the 43 identified profiles, according to table 2 of events. Altitude groups were selected where the decrease in height was greater than or equal to 30% in relation to climatology.

The caption for Figure 8 says '24.1 -28 km' altitude but the plots show a vertical profile extending well above 28 km. The big anomalies are all at very high altitudes – above 30 km – how is this connected to the changes in total ozone?
The legend corresponds to what is observed when analyzing satellite data in ozone partial pressure units, which corresponds to this more pronounced decrease between altitudes of 24.1 - 28 km. According to the figure shown above.

How can this be in any way connected to the Antarctic ozone hole, if the ozone anomaly is above 35 km?
As already mentioned above, the vertical profile analyzes are in ozone mixing ratio units. When these graphs were first created, it was decided to analyze the profiles in ozone partial pressure, which better characterizes the AOH region, at lower altitudes, even considering other constituents. Bresciani et al. (2018) presents a multi-instrumental analysis using vertical profile data on ozone partial pressure, in mid-latitude regions.

I can't understand what is being said in the caption of Figure 9 and it seems inconsistent with Figure 8.
Figure 9 shows the percentage differences of events in relation to climatology for each month from 1979 - 2020, with SABER data analyses. This figure shows the most intense decreases in the months of September and October, in the layer between 24.1 and 28 km, considering the calculation of percentage differences, equation 2 in the manuscript:

$$RD\ (z) = 100 * \frac{OMR_d(z) - \overline{OMR}_m(z)}{\overline{OMR}_m(z)}$$

Its calculation is based on the difference between the OMR of the day of the event ($OMR_d(z)$) and the corresponding multi-year monthly averaged OMR ($\overline{OMR}_m(z)$).

Figures 10 and 11 are clear.
Okay.

For Figure 12 I believe from the caption you've plotted the climatology of the zonal wind for August to November and not the anomaly for your identified events?

Yes, correct, it's just the climatology all the period data analysis (1979-2020).

The caption for Figure 13 says it's the average for the identified events but doesn't look very different to Figure 12. Regardless of whether I have understood the captions correctly, the essential point is you need to show the difference in what the zonal wind looks like when there is a low ozone event in Santa Maria compared to when there is not.

Figure 12 presents the monthly climatological behavior for the entire period of data analyzed with reanalysis (1979 - 2020), to observe the zonal behavior of the wind, position, and intensity of the jet streams. On the other hand, figure 13, despite presenting the climatological average, represents the average zonal and jet stream behavior for AOH-influenced events in the SM region during the same period.

Figure 14 doesn't match figure 13 at all which confuses me greatly.
In figure 13 there is only one tropospheric jet and it is centered on a latitude of about 35 degrees. Santa Maria appears on the equatorward edge of this jet.
In figure 14 there are two distinct tropospheric jets centered on approximately 55 degrees and 25 degrees. The "conceptual model" seems to have no connection to the earlier results.

Figure 13 presents, based on the results obtained with ECMWF ERA5 data, the average behavior of the zonal wind during the recording of AOH influence events over mid-latitude regions. It is observed that the jets connect, on average, during these events, which dynamically explains how masses of air poor in $O_3$ reach regions of medium latitudes, see case studies presented in the manuscript, and the other 100 events identified from the analysis of the total ozone column for Santa Maria.
The representation of the conceptual model considers the standard behavior of the atmosphere, where two tropospheric jet streams (polar and subtropical) and the stratospheric jet stream, known as the polar vortex, are observed.

Additionally, Figure 14 shows ozone poor air ending up at subtropical latitudes between 700 and 500 hPa, that is, the middle troposphere. This seems completely unconnected to the rest of the work. This point was raised in the initial review but has not been addressed by the authors.

The representation of the conceptual model has been changed, and the new version presents a model between 500 and 5 hPa. Since the other analyses, climatology, and case studies, show similar behavior to what was presented in the average of all identified events (figure 13) and, for the entire data period (1979 - 2020), through the analyzes carried out with the ECMWF ERA5 reanalysis.

---

## Author Comment (AC2)

*We would like to thank the second referee for his time, valuable feedback, and comments. The authors' comments and response (in red), and the changes will be in the new version of the manuscript (in red).*

The authors have addressed some of the issues I raised in my earlier review (I was reviewer #2 last time round). However, I fear there is still more work to be done to make this paper ready for publication. The bulk of the work needed amounts to providing clearer and more complete description of what is being presented. I describe these needs point by point below.

**High level comments:**

Aside from those more minor issue, I still have some higher level concerns. First it is still somewhat unclear what specifically is new in this paper. The manuscript still frequently interjects references to previous work by the author and others, making it hard to discern the specific new insights being gained here.

Additionally, I remain highly skeptical about details of the presented conceptual model. In terms of vertical profiles of ozone, your own analysis (which only considers altitudes above 22km) shows that the largest impact of these events is in the 22-26km region. How then, does your conceptual model have blue lines showing transport of negative ozone anomalies down to ~800 hPa. You present no evidence for transport to these depths. Such conceptual model would need to be backed up by Lagrangian trajectories or some kind of "tagged ozone" model study. As you've drawn it, how does your model explain what is seen in the SABER data? Your blue lines don't go anywhere near 20 km in at the latitude of your observations. You really need to rethink this model, particularly if, as you say in your response to the earlier reviews" this is the point that is "new" in your study.

Many others have looked such events, which have been termed "ozone mini holes". I suggest you read Millan and Manney, 2017 (doi:10.5194/acp-17-9277-2017) and the many references therein and see how your analysis agrees with theirs, and what your studies add. Also look at more recent papers that cite them or others (it's been a while since I was familiar with the work of those authors).

**More specific comments:**

Abstract: It feels odd that you don't mention the use of SABER data in the abstract, particularly if (as seems possible), this is the "new" aspect to the work. Please consider working that in.
Okay. The abstract was rewritten.

Line 34: "barrier in" -> "barrier to"
Okay.

Line 35: The first open parenthesis is never closed. I don't think you want it here in any case. Also: "polar region" -> "polar regions" and "medium latitudes" is better as "midlatitudes".
Okay.

Line 62: Insert "from" after "software"
Okay.

Line 65: "validated" -> "valid"
Okay.

Line 75: "Brewers" -> "Brewer"
Okay.

Line 76: Delete "published by"

Okay.

Line 85: "replaced" -> "succeeded"
Okay.

Line 87: "AURA" -> "Aura"
Okay.

Line 88: You've already defined NASA above, no need to define it again., just say NASA.
Okay.

Line 89/90: Similarly, you already defined TOMS above.
Okay.

Line 91: "satellites" -> "satellite instruments"
Okay.

Line 93: Insert "spectral" between "740" and "channels". Perhaps also discuss the number of pixels across the track too.
Okay.

Line 100: "analysis" -> "analyses"
Okay.

Line 103: "varies from … to …" -> "switches from … to …" or "alternates between … and …".
Okay.

Line 107: Delete "of" before "altitudes"
Okay.

Line 114: If you're going to stick to this bizarrely fine grid, please at least note the original SABER resolution in this discussion.
Okay.

Line 121: "regarding" -> "focusing on"
Okay, the sentence has been rewritten.

Line 133/134: Change sentence to: "In particular, they enable identification of the origins of air-masses (Holton, 1995). In the Southern Hemisphere (SH) Absolute Potential Vorticity (APV) is used.". However, I'm going to suggest that you eliminate all discussion of APV and instead simply discuss PV, changing your adjectives from low, to high and vice versa, etc. This is because, in none of your figures do you plot APV, you instead consistently plot PV. Why plot one thing and talk about another. It's not hard for the reader to flip the sign in their heads, but why make them do the unnecessary work.
The sentence was rearranged, removing the explanations about APV. Throughout the text only PV will be used.

Line 136: Insert "the" before "AOH" I'd say
Okay, the paragraph has been rewritten.

Line 138: "preferred" seems an odd choice of word how about "the height at which the largest decreases in ozone mixing ratio occurred, as discerned from SABER profiles"?
Okay, the paragraph has been rewritten.

Line 150: I suggest "… presents values lower than the corresponding multi-year monthly means (TCOm) minus 1.5 times the corresponding standard deviation…"

Okay.
Line 150-153: I'm not clear what is being said here. Does Peres describe an alternative approach that you're contrasting yours with, or does that paper lay the theoretical basis for the approach you're describing above (in which case, cite it in the first sentence in this subsection). In any case, this sentence interrupts the flow where you talk about your definition of a AOH event.

Also, lines 150-155+: You basically define this criteria three times, once in line 150, the second on line 155, then again in the equation immediately below. Surely once (or at most twice) is enough. Please condense this discussion.

The paragraph (150-155+) has been reorganized for better reader understanding.

Line 163 (first line of section 2.2.1): "on 42 years" -> "in 42 years"
Okay.

Line 165: "on the platform" -> "from SABER"
Okay.

Line 166: "the occurrence of an" -> "each"
Okay.

Line 169: insert "multi-year" before "monthly"
Okay.

Line 187: "dynamic" -> "dynamics"
Okay.

Section 3.3.1 title: Insert "of" after "event".
Okay.

Line 215/216: This sentence feels out of place. The one before and one after talk about the 2016 event, the one after does too. Why interrupt with a reference to a different event here?
Okay. Sentence removed.

Line 223: Insert "a" before 23%. Also "into" after "continued"
Okay.

Line 237 (Section title): Shouldn't it be 3.3.2, not 3.2.2?
Okay.

Lines 244: Again, why talk about APV when the plot is PV (so sign flipped). I'd talk about PV decreasing rather than APV increasing. (Or you could make the plot be of APV, but I vastly prefer the former, after all, it's not like you're talking about "Absolute latitude").
Okay.

Lines 249-251: You use terms like "under" and "over" here, but it's not clear whether you mean vertically (as in "high ozone over Brazil") or north/south. Please say "north of", "south of" etc. if that's what you mean. If you really do mean "over", "under" in the vertical sense, how is the reader supposed to deduce that from your maps, which are at a single level. If you're referring to other maps not shown, then say so.
The sentence was re-written and re-arranged.

Lines 252-269: This whole discussion seems to not be about the "case study". I still don't get why it is here. Also, once again, having this here makes it very hard to discern what is "new" in this study. I presume we resume "new" stuff at line 271, but having this long narrative in the middle of the case-study section feels disjoint.

Okay. Paragraph removed.

Line 270, discussion of Figure 6: Nowhere in the discussion or the caption of Figure 6 do you tell us what longitudes this plot corresponds to. Is it a cross section over Santa Maria? An average over some span of latitudes in the Brazil region? A zonal mean. This is a key point in your discussion, but the reader is not able to understand it.

The figure represents a vertical section of the atmosphere over the longitude of Santa Maria (53ºW). The triangle shown in the figure represents the latitude, on the x axis, of Santa Maria (29ºS). The text will be re-written.

Line 276: "between 200 hPa" and what, do you mean "at 200 hPa" or between 200 and some other level.
Okay. Removed "between"

Line 280: Be specific on "high levels."
Okay.

Line 293-294: This sentence is badly worded, it sounds like the analysis took 42 years to perform, not that your analyzing 42 years' worth of data.
Okay, inserted in the text "of data analyzed".

Lines 313-318: I am confused as to what is being described here and presented in Figure 7. You talk about a histogram of "events" at different levels, but you have not defined what you mean by an "event" for the SABER data. The discussion in section 2.2.1 established your "RD(z)" metric, but you never discussed any kind of "threshold" for it, nor do you refer back to this definition in the discussion here. Are you using the RD metric? If so, what is the threshold for an "event" that appears in the bar chart. If you're not using your RD metric, why did you take the time to define it?

Correct, figure 7 are analyzes of the vertical profiles available for the dates of AOH influence events identified, through the methodology presented in section 2.2, for Santa Maria. Reductions greater than or equal to 15% in relation to the monthly climatology for the period of available SABER data are analyzed. Visually, the 43 event profiles found (SABER data for the same date as the identified event, via table 2) were separated into groups where reductions greater than 15% were considered.
The text in the new version of the manuscript has been rewritten.

Line 334/335: This sentence is very unclear. It sounds like you're saying that the average number of events is reduced more intensely. I think you mean to say something like: "… stand out, with more intense reductions in the ozone mixing ratios averaged over the events compared to the climatology than in other months".
Okay.

Line 349: Suggest "increase in absolute potential vorticity" change to "decrease in potential vorticity"
Okay.

Line 352: Insert "e.g.," before the Bittencourt reference.
Okay.

Line 354/244: "This value has had the monthly climatological … months of August and November subtracted"
Okay. Sentence removed.

Lines 355-360: Here you've really tied yourself in a knot over the APV thing. You talk about negative PV anomalies standing out, and all the numbers you quote are negative, but you still persist in pretending you're talking about APV. Give up on the "absolute" thing and just talk about PV the way you talk about latitude, there is no shame in negative numbers!

Okay. Decrease.

Line 362: "to understand" -> "for understanding"
Okay.

Line 363: insert "in" after "important"
Okay.

Line 368: Once again, nowhere do you tell us what longitude(s) this plot is for. This is (probably) a critical plot to your argument, but you don't help the reader understand it.
Okay. The sentence was rewritten.

Line 387: "Negative temperature"! Didn't I comment on that last time. Say low temperature or say negative anomalies (but compared to what, if it happens each year it's not an anomaly). Fundamentally, I am still very unhappy about your conceptual model.
Okay. The phrase was removed and reorganized.

Line 415: "… Santa Maria, a midlatitude region, during the…"
Okay.

Line 438/439: Again, give up on APV and just talk about PV
Okay.

Figure 2: As requested before, please artificially stagger the different lines in the x-direction, so we can more readily distinguish the error bars.

Figure 4: Get the fact that this is profiles/averages over your measurement location into the caption (give latitude/longitude range of "box")
Okay.

Figure 6: Again, you do not tell us the longitude (or longitude range or that it is a zonal mean) for this figure. Also, you don't define the feint gray lines with arrows on them.
Okay.

Figure 8: This caption is confusingly worded. "Average of events per month" sounds like a number of events not an ozone mixing ratio should be something like "average ozone mixing ratios for the identified events … compared to the monthly climatology …."
Okay. Thanks for the suggestion.

Figure 9: Similarly, "average the AOH influence events per month" sounds like you're counting events. Say instead something like "Figure 9: Average impacts of AOH-influence events for (a) August, b) September, c) October and d) November from 2002 to 2020, as measured by SABER. Lines show differences between the averages of all profiles influenced by AOH events each month and the corresponding monthly climatology, expressed as percentage of the climatology."
Okay. Thanks for the suggestion.

Figure 12: Again, talk about the longitude (or range or zonal mean) and define the light grey lines.
Okay.

Figure 14: See discussion at the top of this review.
Okay.

---

## Author Response (AR1)

Review of "Multi-instrumental analysis of ozone vertical profile and total column in South America: comparison between subtropical and equatorial latitudes"
Gabriela Dornelles Bittencourt et al.

**Summary and General Comments:**
This paper examines long time series of vertical profile and total column data to describe the differences in behavior of ozone between a subtropical station (Santa Maria) and tropical station (Natal) in Brazil. The authors leverage the SABER instrument (vertical profile; both stations) and TOMS and OMI (total column; both stations), Ozonesondes profiles (Natal), Dobson (Natal), and Brewer (Santa Maria) data. The authors validate the SABER profile data with Natal to justify its use to demonstrate the vertical profile differences between the two stations, compare TOMS and OMI to the ground-based total ozone instruments, and perform a wavelet analysis to show the effect of climate and solar oscillations on the total column and lower stratospheric ozone data.
The analyses and conclusions drawn from them are generally sound. However, the text needs a major reworking. The authors should carefully go through the paper and clean up the writing for better clarity. There are also several sections, particularly in the Introduction, that are highly repetitive. There are many times where very similar explanations of the Brewer-Dobson Circulation are invoked, for example.
A note on data availability: It is not made clear if all the data are publicly available and where the data sets are located. I am particularly interested in locating the Santa Maria ground-based total column ozone data. Please indicate in a Data Availability section where all the data used in this paper are located, and if they are not currently publicly available, they should be made so.

**Recommendation:**
While I do not find many problems with the analysis and technical details of the manuscript, the text requires substantial edits. A second version of this paper may become acceptable for publication.
The manuscript has been substantially edited and revised as suggested by the reviewer to improve English and clarity. Major edits are highlighted in red in the revised manuscript.

**Specific and Line-by-Line Comments:**
Please add the SHADOZ v6 Ozonesondes data Doi where appropriate (e.g., Line 137) and in the reference list: https://doi.org/10.57721/SHADOZ-V06
The SHADOZ data DOI was added as per the request of the reviewer.

Line 71: Natal has Ozonesondes data dating back to 1979 on WOUDC (before SHADOZ).
Yes, Natal has Ozonesondes data dating back to 1979. However, this ozone time series is not continuous and has significant periods of gaps. The corresponding paragraph was reworded as follows in the revised manuscript: Due to the limited number of radiosondes carried out at SM, a comparison between radiosondes and SABER was only made for the equatorial site Natal. Natal is one of the oldest ozone stations in South America with a record of ozone profiles dating back to 1979, although there are large gaps in the data. The most continuous ozone profile experiment in Natal is a weekly-based one, which started in 1998 in the SHADOZ framework.

Line 85: You are referring specifically to the total column ozone measurements here, correct? Please clarify.
It is correct. The sentence is changed as follows: These two locations are the most important stations in Brazil with ground-based **total ozone** measurements which began in 1992 for SM and in 1994.

Lines 121-125: These two sentences are not necessary and are another example of the repetitive nature of the text.
Removed as per the request of the reviewer.

Line 139: Based on Figure 2, it looks like data only up through 2018, not 2020, have been used.
We thank the reviewer for this relevant observation. The analysis period covers 1998-2020. There was a clerical error in creating Figure 2, which stopped in 2018. The figure has been redrawn in the revised version to show ozone mixing ratio vertical distributions over the whole period, from 1998 to 2000.

Line 139: What do you mean by "604 vertical levels?" Were the Ozonesondes profiles averaged into altitude bins? If so, why 604?
For our study, ozone and temperature profiles obtained from balloon-sonde were vertically interpolated from the surface up to ~30km altitude, with a resolution of 50m, which gives 604 altitude bins. This was clarified in the revised manuscript.

Lines 142-143: "Atmosphere Survey using Broadband Emission Radiometry" for the SABER acronym is incorrect, but it was correctly defined earlier in the paper, so this can be deleted.
Deleted as pointed out by the reviewer.

Line 163: Please use commas for "6,715" and "3,681" instead of "6.715" and "3.681"
Amended.

Line 183: The Aura satellite, not ERS-2.
Amended in the revised manuscript.

Section 2.2: I think this section can be removed. No need to spend time explaining these widely used statistical measures. I recommend keeping Section 2.2.1, however. If the decision is made to keep this Section, the Equations 1, 2, and 3 should have "SATELLITE" not "SATELITE"
Amended in the revised manuscript.

Figure 2: Similar to how Figures 3 and 4 were constructed, it would be very illuminating to see a third panel on Figure 2 showing the full time series of differences in ozone mixing ratio for coincident measurements from the two instruments (Ozonesondes and SABER).
Following the reviewer's recommendation, figure 2 has been modified.

Lines 216 and 256: Better to use "coincident" than "concomitant." As previously stated, the paper would benefit greatly from a thorough editing of the text.
Amended in the revised manuscript.

Line 248: Data gaps should not really affect the comparisons of coincident satellite and Ozonesondes data.
The sentence is amended accordingly.

Figures 3, 4, 5, 7, and 9: What do the error bars represent? One standard deviation?
Yes, error bars represent ±1s (standard deviation). This is added in the figure captions of Figures 3, 4, 5, 7, and 9.

Line 319: 45N and 45-60S are mid-latitudes, not polar latitudes.
Amended in the revised manuscript.

Figure 8: There are some typos in the titles of these figure panels.
Amended in the revised manuscript.

Figure 8 c and d: The Natal TCO time series do not appear to be daily observations. Please check.
We thank the reviewer for this relevant observation. Figures 8a and b depict daily TCO values (in DU) from Brewer, TOMS and OMI observations, respectively, at SM, over the 1979-2020 period, while Figures 8c, and d show the monthly TCO averaged values from Dobson, TOMS and OMI at Natal. The caption of Figure 8 was amended accordingly.

Table 1: What are the units of the 0.34, -0.07, -0.41, and -0.66 values? It looks like percent, but please make that clear.

It is in per cent (%). For clarity, the table was changed.

Line 420: I'm not familiar with the "influence cone" for this wavelet analysis. Please provide a brief explanation as part of Section 2.2.1. I will say that Figures 10 and 11 are a nice demonstration of QBO and solar cycle oscillations on ozone amounts over the two stations.

The cone of influence is a numerical parameter used in wavelet analysis to define the region of the spectrum which should be considered in the analyses with confidence. It indicates areas where edge effects occur in the analyzed time series. This has been added to the revised manuscript.

The following is a reviewer's report for "Multi-instrumental analysis of ozone vertical profile and total column in South America: comparison between subtropical and equatorial latitudes"
Gabriela Dornelles Bittencourt et al.

1. The introduction needs to be rewritten/restructured. The current form completely diluted the focus of this paper. The authors should rather consider a discussion of irregular distribution of global Ozonesondes stations (Tarasick et al., 2019), how the Santa Maria Ozonesondes can fill the observational gap in South America, and then emphasis that these data are an important addition to understand tropospheric ozone & air quality in a region where scientists are less studied before.

The co-authors would like to thank the reviewer for his comment. The introduction has been restructured and the main objectives reworded in line with his/her suggestions.

2. 163, does it mean that SM has a nearly doubled sampling frequency than NT? The SHADOZ network typically has once per week or fewer sampling frequency, if SM has a greater data coverage, the authors should explicitly state that SM dataset is expected to have a greater statistical power for analysis (e.g. Jaffe & Ray, 2007).

This concerns SABER profiles only, not SHADOZ profiles. Given the quasi-polar orbit of the TIMED satellite and the difference in latitude between SM and NT sites, one expects to obtain more coincidences for the SM site than for NT. This means that the OMR profiles obtained by SABER over SM are expected to have a greater statistical power for analysis than those for NT (Jaffe & Ray, 2007). This was stated in the revised manuscript.

3. Figure 3: it is not very informative for ozone variability, I would like to see all individual profiles at SM plotted in thin gray from the background at each corresponding month (e.g. Fig 2 of Jaffe et al., 2018). Same for Fig 4 & 5.

Figures 3 and 4 show the validation of satellite data through the SHADOZ network from the Natal station, equatorial region of South America. The objective in these analyzes is to show whether there is agreement between the two databases, for the same station. Furthermore, it is observed that satellite measurements below 20 km present many disagreements, with relative differences greater than 50% compared to SHADOZ data. After that, it was observed that, in the lower stratosphere, the two databases agree well with each other. Figure 5 then shows the comparison of Santa Maria and Natal with data from the SABER satellite, to show that this type of analysis, for mid-latitude regions, well identifies the vertical behavior of ozone.

4. Unlike tropospheric ozone, TCO and stratospheric ozone tend to be steadier and latitude-dependent, does this study suggest that the current SHADOZ network is sufficient to monitor tropical stratosphere, or additional stations, such as SM, are also desirable?

We agree with the reviewer that both TCO and stratospheric ozone levels tend to remain steadier and vary with latitude. This highlights the significance of establishing stratospheric ozone measurement stations in regions beyond equatorial and tropical latitudes, especially in the southern hemisphere. The SM site is an excellent location to participate in this mission of monitoring and observing stratospheric ozone since the site often experiences AOH events. Additionally, there are no long-term ozone profiling stations at this latitude or nearby in the SH. A short paragraph has been included in the conclusion to address the issue.

5. Data availability section is a requirement for AMT.
Data availability information has been added, as per the request of the reviewer.

6. I found it is unpleasant to read and grammar edits/checks are required throughout the paper. There are too many to identify in the minor comments below.

We thank the reviewer for his relevant comments, suggestions, and edits in improving our paper. The manuscript has been checked and revised to improve English and clarity. Major edits are highlighted in red in the revised manuscript.

l.205 "root"
Amended.

l.207 delete 'also called residuals', e.g. MAE is also a type of residuals
Amended.

l.221 'Variabilities the ozone' is not English
Amended.

L.389 'interannual'
Amended.

Jaffe, D. A., Cooper, O. R., Fiore, A. M., Henderson, B. H., Tonnesen, G. S., Russell, A. G., ... & Moore, T. (2018). Scientific assessment of background ozone over the US: Implications for air quality management. Elem Sci Anth, 6, 56.

Jaffe, D., & Ray, J. (2007). Increase in surface ozone at rural sites in the western US. Atmospheric Environment, 41(26), 5452-5463.

Tarasick, D., Galbally, I. E., Cooper, O. R., Schultz, M. G., Ancellet, G., Leblanc, T., ... & Neu, J. L. (2019). Tropospheric Ozone Assessment Report: Tropospheric ozone from 1877 to 2016, observed levels, trends, and uncertainties. Elem Sci Anth, 7, 39.

---

## Referee Report (RR1)

Second Review of "Multi-instrumental analysis of ozone vertical profile and total column in South America: comparison between subtropical and equatorial latitudes"

Gabriela Dornelles Bittencourt et al.

This is my second review of this manuscript, and I thank the authors for carefully responding to my first set of comments and for substantially improving the writing in the paper. I support the publication of this paper once my following additional minor and technical comments are addressed.

Minor/Technical Comments:

Line 78: The latitude of SM listed here is different from the abstract. Please check.

Line 81: Suggest deleting the sentence that begins with "It has been operating...". This is already mentioned previously on this page.

Line 84: Confusion on dates. Ground-based monitoring at NT began in 1979, not 1994, correct?

Line 126: Change "southern tropics" to "tropics to sub-tropics"

Line 148: Change to "SABER *measurements* significantly *overestimate* ozone *compared to* the LiDAR or RS"

Line 184: OMI is written as "IMO"

Line 331: Change to "Figure 8 shows the daily (SM) and monthly (NT) TCO values from ground-based and satellite instruments." I am still a bit confused why only monthly values are shown for Natal.

Line 353: OMI again written as "IMO"

Lines 363 and 364: Please make clear that these are percent values (if I am correct)

Table 1: This table seems to have disappeared in my copy of the paper. Please check.

Lines 377-380: I strongly disagree that the annual cycle in TCO from Natal is dominated by the stratosphere. The annual cycle of tropospheric ozone alone from the April-May minimum to the biomass burning enhanced September-October-November season is 20 DU (25 DU minimum to 45 DU maximum). See figure below:

[Figure]

Figure 5 Caption: Typo – "OMI vertical profiles *are* given in ppmv"

---

## Author Response (AR2)

There is still a missing piece between 2.2.1 (wavelet decomposition) and 3.3.2. How can the authors be so sure those are solar and QBO signals? To me, they just use visual inspection of the frequency spectrum. Many previous studies used the wavelets that do not imply it is always appropriate. The authors should have provided some quantifications, e.g., XX% [plus uncertainty] of variability can be attributed to solar cycle, QBO and ENSO, etc.

The authors are thankful to the Referee for this relevant comment that helps to improve the manuscript. We appreciate his/her view that the wavelet analysis method may not be appropriate to quantify the contributions of modes like QBO or ENSO to ozone variability. To this end, and following the Referee's recommendations, a multilinear regression model is used to calculate variability and trend (Trend-Run model). A presentation of the model has been added at the end of the **2.2.1 subsection** (see line 230 in the revised manuscript). The Trend-Run model was applied to the stratospheric OMR time-series obtained over the two study sites (Natal and Santa Maria) from SABER observations at 24km. The percentage contributions of the QBO, ENSO and Solar forcings are presented and discussed at the end of the **3.3.1 subsection** (see line 445 in the revised manuscript).

Second Review of "Multi-instrumental analysis of ozone vertical profile and total column in South America: comparison between subtropical and equatorial latitudes"

Gabriela Dornelles Bittencourt et al.

This is my second review of this manuscript, and I thank the authors for carefully responding to my first set of comments and for substantially improving the writing in the paper. I support the publication of this paper once my following additional minor and technical comments are addressed.

The authors appreciate all the suggestions provided by the reviewer and value the support in enhancing and refining this manuscript.

Minor/Technical Comments:

Line 78: The latitude of SM listed here is different from the abstract. Please check.

Accordingly, the latitude reported in the abstract is incorrect. It is amended in the revised manuscript.

Line 81: Suggest deleting the sentence that begins with "It has been operating…". This is already mentioned previously on this page.
Amended in the revised manuscript.

Line 84: Confusion on dates. Ground-based monitoring at NT began in 1979, not 1994, correct?
Correct. In 1979, measurements began in NT with the Dobson spectrophotometer, and it was only in 1994 that the Brewer spectrophotometer began operating in the region. Amended in the revised manuscript.

Line 126: Change "southern tropics" to "tropics to sub-tropics"

Amended in the revised manuscript.

Line 148: Change to "SABER measurements significantly overestimate ozone compared to the LiDAR or RS."

Amended in the revised manuscript.

Line 184: OMI is written as "IMO"

Amended in the revised manuscript.

Line 331: Change to "Figure 8 shows the daily (SM) and monthly (NT) TCO values from ground-based and satellite instruments." I am still a bit confused why only monthly values are shown for Natal.

Amended in the revised manuscript. The NT TCO database used in the analyzes corresponds to the monthly average. Hence the difference between the sampling in figure 8 between SM and NT.

Line 353: OMI again written as "IMO"

Amended in the revised manuscript.

Lines 363 and 364: Please make clear that these are percent values (if I am correct)

Amended in the revised manuscript.

Table 1: This table seems to have disappeared in my copy of the paper. Please check.

For some reason that I don't know, the table wasn't there in the version I sent. In the new version of the manuscript, it was added.

Lines 377-380: I strongly disagree that the annual cycle in TCO from Natal is dominated by the stratosphere. The annual cycle of tropospheric ozone alone from the April-May minimum to the biomass burning enhanced September-October-November season is 20 DU (25 DU minimum to 45 DU maximum). See figure below:

[Figure]

Figure 5 Caption: Typo – "OMI vertical profiles *are* given in ppmv"

Agreed. The sentence has been rewritten for better understanding.